# Towards Realistic Data Generation for Real-World Super-Resolution

**Long Peng**[1,2]*, **Wenbo Li**[2]†, **Renjing Pei**[2], **Jingjing Ren**[3], **Jiaqi Xu**[4], **Yang Wang**[1,5]†
**Yang Cao**[1], **Zheng-Jun Zha**[1]
[1] USTC, [2] Huawei Noah's Ark Lab, [3] HKUST(GZ), [4] CUHK, [5] Chang'an University
`{longp2001@mail.,ywang120@}ustc.edu.cn,liwenbo50@huawei.com`

## Abstract

Existing image super-resolution (SR) techniques often fail to generalize effectively in complex real-world settings due to the significant divergence between training data and practical scenarios. To address this challenge, previous efforts have either manually simulated intricate physical-based degradations or utilized learning-based techniques, yet these approaches remain inadequate for producing large-scale, realistic, and diverse data simultaneously. In this paper, we introduce a novel Realistic Decoupled Data Generator (RealDGen), an unsupervised learning data generation framework designed for real-world super-resolution. We meticulously develop content and degradation extraction strategies, which are integrated into a novel content-degradation decoupled diffusion model to create realistic low-resolution images from unpaired real LR and HR images. Extensive experiments demonstrate that RealDGen excels in generating large-scale, high-quality paired data that mirrors real-world degradations, significantly advancing the performance of popular SR models on various real-world benchmarks.

## 1 Introduction

Real-world image Super-Resolution (Real SR) is a fundamental problem in image processing, aiming to enhance the resolution and quality of images in real-world scenarios (Chen et al., 2022; Yu et al., 2024; Liu et al., 2023; Zhang et al., 2023c; Sun et al., 2023; Zhang et al., 2024). It has a wide range of applications across various fields, including photography (Chen et al., 2019) and medical imaging (Li et al., 2021), which enhance human visual perception and the robustness of vision systems (Haris et al., 2021; Noor et al., 2019; Gunturk et al., 2003; Chen et al., 2020a). However, traditional bicubic-interpolation-based Real SR methods have proven less effective in complex real-world scenarios due to the significant discrepancy between the bicubic pattern and real degradation (Chen et al., 2022; Liu et al., 2023; Cai et al., 2019; Wang et al., 2020; Chen et al., 2024). Consequently, substantial efforts have been directed towards developing methods for generating more realistic data to improve the generalization ability of Real SR models (Cai et al., 2019; Wei et al., 2020; Zhang et al., 2021; Wang et al., 2021b; Zhang et al., 2023b; Park et al., 2023; Li et al., 2022b; Xiao et al., 2020; Wolf et al., 2021; Luo et al., 2024; Sun & Chen, 2024; Hendrycks & Dietterich, 2019; Deng et al., 2023).

To explore what kind of data contributes most to the SR model's capability, we synthesize different sets of training data using (Elad & Feuer, 1997) and evaluate the performance of FSRCNN (Dong et al., 2016). As shown in Figure 1(a), the red triangles represent test samples of the target domain, while rectangular boxes indicate training sets with different blur kernels and noise levels. The results clearly demonstrate that the closer the training data distribution is to the test data, the better the model's performance. This underscores the importance of designing a method that can adaptively generate accurate data for different target domains. Therefore, an ideal data generation system for Real SR should meet the following criteria: ***I) Large-scale***, to satisfy the extensive data requirements for training deep learning models (Raicu et al., 2008; Maeda, 2020; Sehwag et al., 2022; Wu et al., 2023); ***II) Realistic***, to enable Real SR models to accurately learn the characteristics of real-world degradation (Cai et al., 2019; Wei et al., 2020; Zhang et al., 2021; Ji et al., 2020; Liu et al., 2022);

---

*Work done when the first author interned at Huawei Noah's Ark Lab.
†Wenbo Li and Yang Wang are the corresponding authors.

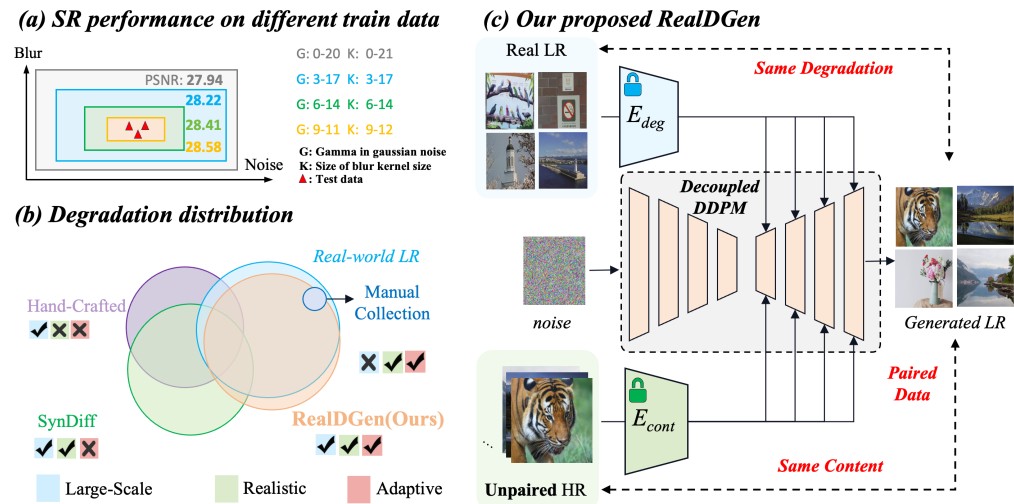

Figure 1: (a) and (b) are SR performance on different train data and degradation distribution of different methods. (c) is the pipeline of our unsupervised data generation framework RealDGen.

and ***III) Adaptive***, to flexibly generate data with arbitrary given degradation patterns, improving generalization in target domains (Liu et al., 2023; Zhang et al., 2023b; Chen et al., 2019; Lugmayr et al., 2019b; Mou et al., 2022).

Existing data generation methods for Real SR (Cai et al., 2019; Wei et al., 2020; Zhang et al., 2021; Wang et al., 2021b; Maeda, 2020; Bulat et al., 2018; Yang et al., 2023; Lugmayr et al., 2020a; Yuan et al., 2018; Ignatov et al., 2017), as shown in Figure 1(b), can be broadly categorized into the following: **a) Manual Collection via Focal Length Adjustment:** This approach involves using digital single-lens reflex cameras (DSLRs) with varying focal lengths to capture images, followed by alignment (Cai et al., 2019; Wei et al., 2020). While it can produce realistic paired data, it is labor-intensive and often results in scene monotony and image misalignment, failing to meet the large-scale data requirement. **b) Hand-Crafted Physical-Based Degradation Modeling:** This method employs various degradation models (*e.g.*, noise, blur, bicubic, JPEG) applied in single-order or higher-order combinations (Zhang et al., 2021; Wang et al., 2021b). Although efficient in generating large data quantities, the synthetic data often fails to accurately reflect the complex degradation patterns of real-world images, and its lack of adaptability to specific domains limits effectiveness. **c) Learning-Based Methods:** Techniques involving Generative Adversarial Networks (GANs) (Bulat et al., 2018) and diffusion models (Yang et al., 2023) are proposed to simulate realistic real-world degradation for low-resolution (LR) images. While these methods produce more realistic data compared to hand-crafted approaches, they often struggle to generalize to new and diverse domains, limiting their applicability in real-world scenarios. In summary, existing data generation methods face challenges in achieving both realism and adaptability: tailoring models to specific target domains may hinder their adaptability to new domains, and vice versa. Overcoming these challenges is crucial for advancing the field of Real SR.

In this paper, we introduce a novel unsupervised learning framework Realistic Decoupled Data Generator (RealDGen) to meet the large-scale, realistic, and adaptive data generation criteria, as shown in Figure 1(c). RealDGen enhances degradation realism and content fidelity by separately modeling content and degradation through unsupervised learning and integrating them into a diffusion model to generate paired data. The training involves two steps: first, pre-training degradation and content extractors using contrastive and reconstruction learning to improve representation robustness; second, using these pre-trained extractors to condition the diffusion model with real LR degradation and content representations. To improve generalization to unknown LR distributions, the partial parameters of the extractors are fine-tuned. During data generation, unpaired HR and real LR images are used to extract content and degradation representations, which are then combined in the diffusion model. This process produces data that marries HR content with arbitrary LR degradation, improving adaptability to new domains. Extensive experiments show that RealDGen outperforms previous methods in generating realistic paired data and enhancing the performance of SR models in real-world scenarios.

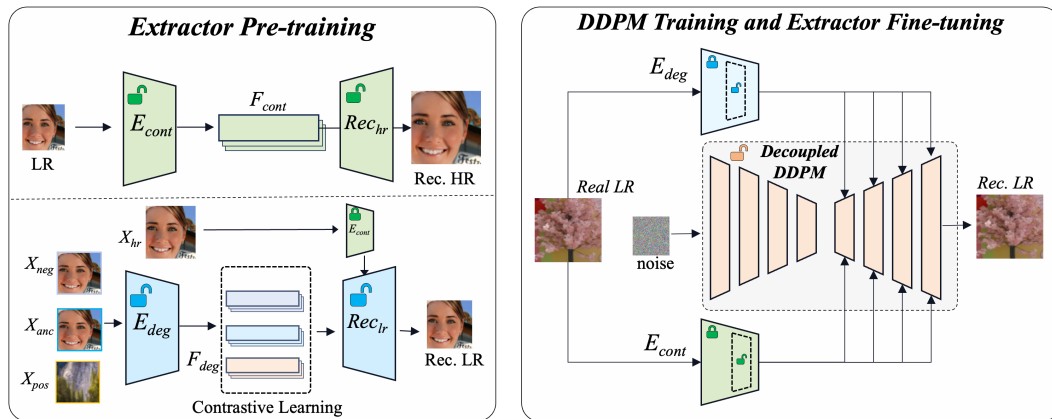

Figure 2: An overview of the training pipeline of our proposed RealDGen. We first train on the content and degradation extractors, then train Decoupled DDPM while fine-tuning the partial parameters of the extractors. RealDGen adaptively generates realistic LR images with arbitrarily given real LR images and unpaired HR images.

The contributions of this paper can be summarized as follows:

- We propose a novel unsupervised Realistic Decoupled Data Generator (RealDGen) to adaptively generate large-scale, realistic, and diverse data for real-world super-resolution.
- We introduce well-designed content and degradation extraction strategies and a novel content-degradation decoupled diffusion model to generate realistic LR with arbitrary unpaired LR and HR conditions.
- Compared with previous methods, our method significantly advances the generalization ability of popular SR models, achieving the best performance on real-world benchmarks.

## 2 METHOD

In this paper, we introduce a novel Realistic Decoupled Data Generator (RealDGen) for real-world super-resolution to adaptively generate large-scale, realistic, and diverse real paired data. In particular, well-designed content and degradation extractor learning strategies are proposed to capture robust content and degradation representations in the real world. A novel content-degradation decoupled diffusion model is proposed to generate realistic LR with arbitrary unpaired LR and HR conditions. The training process is divided into two distinct phases: (a) Content and Degradation Extractor Pre-training and (b) Decoupled DDPM Training and Extractor Fine-tuning, as shown in Figure 2.

### 2.1 CONTENT AND DEGRADATION EXTRACTOR PRE-TRAINING

To capture content and degradation representations, we propose dedicated degradation and content extractors, denoted as $E_{deg}$ and $E_{cont}$, respectively. We employ reconstruction learning for training $E_{cont}$, as shown in the left of Figure 2. Specifically, for a given high-resolution (HR) image $\mathcal{X} \in \mathbb{R}^{C \times H \times W}$, we degrade it to a low-resolution (LR) counterpart $\mathcal{X}_{lr} \in \mathbb{R}^{C \times h \times w}$ by Real-ESRGAN (Wang et al., 2021b) degradation model $\mathcal{D}$ with diverse synthetic degradations. Subsequently, $E_{cont}$ is engaged to extract the content representation $F_{cont}$ from $\mathcal{X}_{lr}$. Thereafter, a HR reconstruction network $Rec_{hr}$ is harnessed to reconstruct the HR image $\hat{\mathcal{X}} \in \mathbb{R}^{C \times H \times W}$ from $F_{cont}$, as follows:

$$F_{cont} = E_{cont}\left(\mathcal{X}_{lr}\right), \hat{\mathcal{X}} = Rec_{hr}\left(F_{cont}\right) .$$ (1)

The objective is to minimize the reconstruction loss $\mathcal{L}_{rh}$ between the reconstructed image $\hat{\mathcal{X}}$ and the original high-resolution image $\mathcal{X}$, as follows:

$$\mathcal{L}_{rh} = \frac{1}{N} \sum_{i=1}^{N} \left(\hat{\mathcal{X}}_i - \mathcal{X}_i\right)^2$$ (2)

| **Algorithm 1** Decoupled DDPM Training | **Algorithm 2** Data Generation |
|---|---|
| 1: **repeat** 
 2:    $\mathbf{x}_{lr} \sim q(\mathbf{x}_{lr})$ 
 3:    $t \sim \text{Uniform}(\{1, \ldots, T\})$ 
 4:    $\boldsymbol{\epsilon} \sim \mathcal{N}(\mathbf{0}, \mathbf{I})$ 
 5:    $F_{cont} = E_{cont}(\mathbf{x}_{lr})$ 
 6:    $F_{deg} = E_{deg}(\mathbf{x}_{lr})$ 
 7:    $\mathbf{c} = \mathcal{M}(F_{cont}, F_{deg})$ 
 8:    Take gradient descent step on 
      $\nabla_\theta \left\| \boldsymbol{\epsilon} - \boldsymbol{\epsilon}_\theta(\sqrt{\bar{\alpha}_t}\mathbf{x}_0 + \sqrt{1 - \bar{\alpha}_t}\boldsymbol{\epsilon}, \mathbf{c}, t) \right\|^2$ 
 9: **until** converged | 1: $\mathbf{x}_{lr} \sim q(\mathbf{x}_{lr}), \mathbf{x}_{hr} \sim p(\mathbf{x}_{hr})$ 
 2: $\mathbf{c} = \mathcal{M}(E_{deg}(\mathbf{x}_{lr}), E_{cont}(\mathbf{x}_{hr}))$ 
 3: $\tau \sim \text{Uniform}(\{1, \ldots, T\})$ 
 4: $\mathbf{x}_t = (\sqrt{\bar{\alpha}_t}\mathcal{D}(\mathbf{x}_{lr}) + \sqrt{1 - \bar{\alpha}_t}\boldsymbol{\epsilon}), \boldsymbol{\epsilon} \sim \mathcal{N}(\mathbf{0}, \mathbf{I})$ 
 5: **for** $t = \tau, \ldots, 1$ **do** 
 6:    $\mathbf{z} \sim \mathcal{N}(\mathbf{0}, \mathbf{I})$ if $t > 1$, else $\mathbf{z} = \mathbf{0}$ 
 7:    $\mathbf{x}_{t-1} = \frac{1}{\sqrt{\alpha_t}}\left(\mathbf{x}_t - \frac{1-\alpha_t}{\sqrt{1-\bar{\alpha}_t}}\boldsymbol{\epsilon}_\theta(\mathbf{x}_t, \mathbf{c}, t)\right) + \sigma_t \mathbf{z}$ 
 8: **end for** 
 9: **return** $\mathbf{x}_0$ |

where $N$ denotes the batch size, which we empirically set to 64. After using LR-HR paired training, $E_{cont}$ is able to learn a robust content representation under diverse degradation and real scenarios.

Considering the variability of degradation in diverse scenarios and imaging devices in the real world, we advocate for contrastive learning (Chen et al., 2020b; Hermans et al., 2017) to curate positive and negative samples for training $E_{deg}$, which guarantees the uniqueness of the degradation representations. Specifically, for a HR image $\mathcal{X} \in \mathbb{R}^{C \times H \times W}$, we generate an LR image $\mathcal{X}_{lr} \in \mathbb{R}^{C \times h \times w}$ by $\mathcal{D}$ with parameter $\theta$ as the anchor $\mathcal{X}_{anc}$. We further obtain a set of negative samples $\mathcal{X}_{neg_i} \in \mathbb{R}^{C \times h \times w}$ by applying $\mathcal{D}$ with different parameters $\theta_i$ to $\mathcal{X}$, and a set of positive samples $\mathcal{X}_{pos_i} \in \mathbb{R}^{C \times h \times w}$ by applying the degradation $\mathcal{D}$ with same parameter $\theta$ to different HR images $\mathcal{X}'_i \in \mathbb{R}^{C \times H \times W}$, as follows:

$$\begin{aligned} \mathcal{X}_{neg} &= \{\mathcal{D}(\mathcal{X}, \theta_1), \mathcal{D}(\mathcal{X}, \theta_2), \ldots, \mathcal{D}(\mathcal{X}, \theta_n)\}, \\ \mathcal{X}_{pos} &= \{\mathcal{D}(\mathcal{X}'_1, \theta), \mathcal{D}(\mathcal{X}'_2, \theta), \ldots, \mathcal{D}(\mathcal{X}'_n, \theta))\}. \end{aligned} \tag{3}$$

The objective is to minimize the contrastive loss $\mathcal{L}_{cl}$ to drive $E_{deg}$ learn the uniqueness of the degradation representations in LR images, suppressing the interruption of content, as follows:

$$\mathcal{L}_{cl} = \sum_{i=1}^n \max\left(0, d\left(E_{deg}(\mathcal{X}_{anc}), E_{deg}(\mathcal{X}_{pos_i})\right) - d\left(E_{deg}(\mathcal{X}_{anc}), E_{deg}(\mathcal{X}_{neg_i})\right) + margin\right). \tag{4}$$

where $d$ symbolizes the L2 distance, $n$ is the number of samples, and we empirically set $n$ and $margin$ to 3 and 0.01, respectively. Furthermore, to drive $E_{deg}$ to learn the complete degradation representations, we utilize the reconstruction learning strategy as aforementioned for supervising $E_{deg}$. Specifically, we utilize the pre-trained $E_{cont}$ to learn the content representation of HR images $\mathcal{X}_{hr}$ and $E_{deg}$ to learn the degradation representations of $\mathcal{X}_{lr}$. We employ a low-resolution reconstruction network, $Rec_{lr}$, to combine these representations and reconstruct the LR image, $\hat{\mathcal{X}}_{lr}$, as follows:

$$\hat{\mathcal{X}}_{lr} = Rec_{lr}(E_{deg}(\mathcal{X}_{lr}), E_{cont}(\mathcal{X}_{hr})). \tag{5}$$

The objective is to minimize the reconstruction loss $\mathcal{L}_{rl}$ to drive $E_{deg}$ learn the completeness of the degradation representations as follows:

$$\mathcal{L}_{rl} = \frac{1}{N}\sum_{i=1}^N \left(\hat{\mathcal{X}}_{lr_i} - \mathcal{X}_{lr_i}\right)^2. \tag{6}$$

After training with well-designed learning strategies, our extractors effectively capture robust content and degradation representations. More details and analysis are provided in Appendix A.1 and A.2.

## 2.2 Decoupled DDPM Training and Extractor Fine-tuning

We introduce a content-degradation Decoupled Diffusion Probabilistic Model (Decoupled DDPM) to generate real LR images. In detail, given real LR images $\mathcal{X}_{lr}$ from real-world $q$ encompassing various degradations, we extract their robust content representation $F_{cont}$ and degradation representation $F_{deg}$ by pre-trained $E_{cont}$ and $E_{deg}$, respectively. To enhance the generalization of $E_{deg}$ and $E_{cont}$ on unseen real distributions, partial parameters are fine-tuned, as shown in Figure 2. A modulation block $\mathcal{M}$ is introduced to adequately incorporate degradation representation into the content, formulated as:

$$\mathbf{c} = \mathcal{M}(E_{deg}(\mathbf{x}_{lr}), E_{cont}(\mathbf{x}_{lr})). \tag{7}$$

Then, this fused image representation is utilized as a condition **c** for controlling our Decouple DDPM to generate LR images. To make it clear, we illustrate the detailed training procedure of the Decouple DDPM, as shown in Algorithm 1. More details of fine-tuning and analysis of $E_{cont}$ and $E_{deg}$ are provided in Sections A.3, A.4 and A.5 of the appendix.

## 2.3 DATA GENERATION

We propose a novel strategy to generate realistic LR images using unpaired LR and HR images by decoupling content and degradation. First, we extract the degradation representation from a real-world LR image and the content representation from an HR image. These representations are combined in the modulation module $\mathcal{M}$ to serve as the condition **c** for the diffusion model to generate LR images. The generated LR images retain the content of the HR image and the degradation of the real LR image, as shown on the right of Figure 1. To enhance fidelity, following (Meng et al., 2021), we denoise from an initial LR image $\mathbf{x}_t$ with $t$ steps of noise rather than from pure Gaussian noise. This initial LR image is degraded by $\mathcal{D}$. In Section 3.4, we analyze the step number $T$. Details of our data generation pipeline are in Algorithm 2.

Although using content and degradation conditions improves the controllability and fidelity of Decoupled DDPM, the inherent stochasticity of the diffusion model (Ho et al., 2020; Rombach et al., 2022) can still introduce tiny artifacts and content distortion. To mitigate this, we propose a filtering mechanism. For each generated LR image, we re-extract content and degradation representations, then calculate the degradation error with the real LR image and the content error with the HR image. By selecting samples with the smallest errors, we reduce diffusion stochasticity and produce higher-fidelity LR images. More details and analysis are presented in Appendix A.3, A.5 and A.6.

## 3 EXPERIMENTS AND ANALYSIS

### 3.1 EXPERIMENTS SETTINGS

**Training details.** We collect about 152,000 real low-resolution images from both public datasets (Wei et al., 2020; Cai et al., 2019; Ignatov et al., 2017) and those captured using smartphones to train RealDGen. Extractors and Decouple DDPM are trained with learning rate $1 \times 10^{-4}$ and batch size 16 on 16 NVIDIA V100 GPUs. More details of the training setting are presented in Section A.1 and A.3 of the appendix. To ensure a fair comparison, we compare our approach with existing methods by using the widely-used DIV2K dataset (Agustsson & Timofte, 2017) as HR images and real LR images as degradation references to create the paired training data for training various popular SR models. To maintain a rigorous and fair comparison, we maintain consistency in all experimental settings and environments, with the exception of data generation. We utilize the public BasicSR for training Real-SR methods with 16 NVIDIA V100 GPUs.

**Compared data generation methods.** We compare our methods with some state-of-the-art real-world data generation methods, including Hand-Crafted Physical-Based Degradation Models (BSR-GAN (Zhang et al., 2021) and Real-ESRGAN (Wang et al., 2021b)) and learning-based degradation diffusion models proposed by (Yang et al., 2023), denoted as SynDiff for convenience.

**Real-SR models.** To comprehensively validate the effectiveness of the generated data, we select five classic and representative backbone architectures for evaluation, including CNN-based model RRDB (Wang et al., 2018), transformer-based model SwinIR (Liang et al., 2021) and HAT (Chen et al., 2023b), diffusion-based model ResShift (Yue et al., 2024) and lightweight model SwinIR-L (Liang et al., 2021). To be fair, we conduct comparative evaluations under consistent experimental conditions and settings. We utilize L1Loss (Chen et al., 2023b; Liang et al., 2021) and perception GAN loss (Wang et al., 2021b; Johnson et al., 2016; Wang et al., 2018) for training PNSR-oriented and Perception-oriented Real SR models, respectively.

**Metrics.** For the PNSR-oriented Real SR model, we adopt PSNR (Huynh-Thu & Ghanbari, 2008) and SSIM (Wang et al., 2004) to quantitatively evaluate the performance. For the perception-oriented Real-SR model, we adopt LPIPS (Zhang et al., 2018) and FID (Heusel et al., 2017) to quantitatively evaluate the performance. DISTS (Ding et al., 2020) and CLIP-Score (Radford et al., 2021) are further introduced to evaluate the accuracy of generated data. Note that the higher the PSNR, SSIM, and CLIP-Score, the better, and the lower the LPIPS, FID, and DISTS, the better.

Table 1: Quantitative comparisons of PSNR-oriented and Perceptual-oriented training SR models on three real-world image super-resolution benchmarks. The best results are highlighted in **bold**.

| PSNR-oriented Training | RealSR | | DRealSR | | SmartPhone | |
|---|---|---|---|---|---|---|
| | PSNR↑ | SSIM↑ | PSNR↑ | SSIM↑ | PSNR↑ | SSIM↑ |
| SwinIR (Real-ESRGAN) | 24.395 | 0.7760 | 26.944 | 0.8308 | 27.395 | 0.8338 |
| SwinIR (BSRGAN) | 25.852 | 0.7808 | 27.985 | 0.8308 | 28.049 | 0.8407 |
| SwinIR (SynDiff) | 25.589 | 0.7687 | 28.301 | 0.8309 | 28.566 | 0.8453 |
| SwinIR (Ours) | **26.094** | **0.7822** | **28.721** | **0.8341** | **28.737** | **0.8489** |
| RRDB (Real-ESRGAN) | 24.579 | 0.7614 | 27.131 | 0.8193 | 27.841 | 0.8378 |
| RRDB (BSRGAN) | 25.406 | 0.7685 | 27.523 | 0.8017 | 28.029 | 0.8278 |
| RRDB (SynDiff) | 25.488 | 0.7691 | 28.078 | 0.8257 | 28.303 | 0.8426 |
| RRDB (Ours) | **26.238** | **0.7747** | **28.727** | **0.8340** | **28.754** | **0.8507** |
| HAT (Real-ESRGAN) | 24.893 | 0.7726 | 27.339 | 0.8215 | 27.781 | 0.8336 |
| HAT (BSRGAN) | 25.997 | 0.7816 | 28.135 | 0.8273 | 28.137 | 0.8369 |
| HAT (SynDiff) | 25.790 | 0.7584 | 28.506 | 0.8286 | 28.508 | 0.8471 |
| HAT (Ours) | **26.140** | **0.7832** | **28.802** | **0.8345** | **28.767** | **0.8489** |
| SwinIR-L (Real-ESRGAN) | 24.367 | 0.7723 | 27.018 | 0.8244 | 27.581 | 0.8409 |
| SwinIR-L (BSRGAN) | 25.651 | 0.7800 | 27.813 | 0.8301 | 28.118 | 0.8437 |
| SwinIR-L (SynDiff) | 25.281 | 0.7516 | 28.170 | 0.8244 | 28.474 | 0.8487 |
| SwinIR-L (Ours) | **26.025** | **0.7810** | **28.869** | **0.8328** | **28.868** | **0.8522** |

| Perceptual-oriented Training | RealSR | | DRealSR | | SmartPhone | |
|---|---|---|---|---|---|---|
| | LPIPS↓ | FID↓ | LPIPS↓ | FID↓ | LPIPS↓ | FID↓ |
| SwinIR (Real-ESRGAN) | 0.3037 | 69.965 | 0.3219 | 39.175 | 0.4053 | 78.242 |
| SwinIR (BSRGAN) | 0.2945 | 79.833 | 0.3023 | 38.541 | 0.3043 | 76.871 |
| SwinIR (SynDiff) | 0.3835 | 103.179 | 0.3801 | 54.588 | 0.3129 | 83.485 |
| SwinIR (Ours) | **0.2536** | **69.736** | **0.2660** | **38.257** | **0.2964** | **74.778** |
| RRDB (Real-ESRGAN) | 0.3480 | 82.056 | 0.3551 | 39.310 | 0.3480 | 77.573 |
| RRDB (BSRGAN) | 0.3041 | 77.412 | 0.3127 | 36.528 | 0.3381 | 78.812 |
| RRDB (SynDiff) | 0.4004 | 98.798 | 0.4017 | 56.573 | 0.3511 | 88.431 |
| RRDB (Ours) | **0.2972** | **76.973** | **0.3077** | **36.259** | **0.3125** | **76.723** |
| HAT (Real-ESRGAN) | 0.3066 | 79.209 | 0.3219 | 41.862 | 0.4022 | 87.950 |
| HAT (BSRGAN) | 0.2852 | 80.192 | 0.2835 | 41.723 | 0.3049 | 81.247 |
| HAT (SynDiff) | 0.3332 | 93.763 | 0.3465 | 50.808 | 0.3171 | 85.587 |
| HAT (Ours) | **0.2457** | **67.573** | **6.2587** | **41.319** | **0.2816** | **76.873** |
| SwinIR-L (Real-ESRGAN) | 0.3108 | 79.491 | 0.3234 | 42.986 | 0.4021 | 87.531 |
| SwinIR-L (BSRGAN) | 0.3013 | 84.195 | 0.2978 | 43.246 | 0.3146 | 83.444 |
| SwinIR-L (SynDiff) | 0.3793 | 98.646 | 0.3748 | 52.464 | 0.3190 | 80.708 |
| SwinIR-L (Ours) | **0.2795** | **75.779** | **0.2862** | **42.542** | **0.3047** | **78.632** |

**Evaluation.** We utilize two public benchmarks to evaluate the performance of real-world image super-resolution methods, including the real-world dataset RealSR (Cai et al., 2019) and DRealSR (Wei et al., 2020) captured by digital single-lens reflex cameras (DSLRs). To further improve the diversity and quantity of real degradation scenarios, we have collected 891 pairs of data captured by smartphones for evaluation, denoted as SmartPhone.

## 3.2 QUANTITATIVE RESULTS

**Generalization ability of Real SR models.** We compare our method with the existing data generation methods on both PSNR-oriented and Perceptual-oriented Real SR models to validate the superiority of our method in boosting the generalization capabilities for real-world image super-resolution, and the results are shown in Table 1. We can observe that our method comprehensively improves the performance of PSNR-oriented and Perceptual-oriented SR models across three benchmarks. It's worth noting that our approach achieves a significant performance improvement, including 0.75 dB in RRDB (Wang et al., 2018) on the PSNR of the RealSR benchmark; 0.296dB and 0.699 dB in

Table 2: Performance comparison of the diffusion-based ResShift model trained on our generated data versus Real-ESRGAN's simulated data on the SmartPhone and DRealSR benchmarks.

| Benchmark | Methods | PSNR↑ | SSIM↑ | LPIPS↓ | CLIP-IQA↑ |
|---|---|---|---|---|---|
| SmartPhone | ResShift (Yue et al., 2024) | 27.05 | 0.806 | 0.352 | 0.546 |
| | **ResShift (Ours)** | **27.27** | **0.818** | **0.345** | **0.557** |
| DRealSR | ResShift (Yue et al., 2024) | 26.19 | 0.755 | 0.413 | 0.574 |
| | **ResShift (Ours)** | **26.32** | **0.772** | **0.378** | **0.622** |

Figure 3: Visual comparison of generated LR. Our method achieves the best visual results with realistic degradation and high fidelity. Please zoom in for better visualization.

SwinIR (Liang et al., 2021) and light-weight SwinIR-L on the PSNR of the RealSR benchmark. Furthermore, our methods also significantly improve LPIPS and FID in four SR models, including 0.0395 and 12.619 in HAT (Chen et al., 2023b) on the LPIPS and FID of the RealSR benchmark. Furthermore, we also conduct experiments on the diffusion-based model ResShift (Yue et al., 2024). Specifically, the official implementation of ResShift is trained on a large-scale dataset using Real-ESRGAN degradation with 500,000 iterations, and we perform a quick fine-tuning with 10,000 iterations using our generated data. As shown Table 2, our proposed method effectively generates accurate and realistic data in the target domain and helps ResShift quickly adapt to the new domain, yielding consistent improvements on both DRealSR and SmartPhone benchmarks. The experiments demonstrate that our method significantly improves performance across various SR approaches, including CNN-based, Transformer-based, and diffusion-based methods.

**Accuracy of generated real LR.** To validate our method's superiority in generating accurately realistic and matched real LR, we conduct comparisons with existing methods in terms of the accuracy of the generated data on six metrics for evaluation. Specifically, we utilize HR images in the three real-world datasets as the content reference and employ the existing methods to synthesize generated LR images, while our method is able to utilize the degradation reference from real LR. As shown in Table 3, we can observe that our method achieves the best performance on three datasets

Table 3: Quantitative comparisons of the accuracy of generating real LR images on three real-world benchmarks under six distinct similarity metrics.

| Benchmark | Methods | PSNR↑ | SSIM↑ | LPIPS↓ | FID↓ | DISTS↓ | CLIP-Score↑ |
|---|---|---|---|---|---|---|---|
| RealSR | Real-ESRGAN | 24.3974 | 0.6798 | 0.3881 | 157.5237 | 0.2618 | 0.7552 |
| | BSRGAN | 23.2665 | 0.6043 | 0.5093 | 177.9996 | 0.3048 | 0.6783 |
| | SynDiff | 25.2461 | 0.7588 | 0.2371 | 119.9950 | 0.1944 | 0.7984 |
| | Ours | **26.1615** | **0.7940** | **0.2226** | **107.2375** | **0.1865** | **0.8238** |
| DRealSR | RealESRGAN | 26.3510 | 0.6935 | 0.3842 | 60.2665 | 0.2437 | 0.7710 |
| | BSRGAN | 26.1970 | 0.6869 | 0.4319 | 62.9189 | 0.2646 | 0.7137 |
| | SynDiff | 28.0125 | 0.8136 | 0.1739 | 31.0855 | 0.1673 | 0.8462 |
| | Ours | **28.6629** | **0.8360** | **0.1497** | **30.0078** | **0.1567** | **0.8577** |
| SmartPhone | RealESRGAN | 26.0680 | 0.5981 | 0.5202 | 106.3958 | 0.2855 | 0.7231 |
| | BSRGAN | 27.6215 | 0.7179 | 0.4399 | 105.3289 | 0.2760 | 0.6771 |
| | SynDiff | 28.5020 | 0.8075 | 0.2423 | 70.6379 | 0.2044 | 0.8213 |
| | Ours | **29.0054** | **0.8292** | **0.2121** | **69.1162** | **0.1964** | **0.8334** |

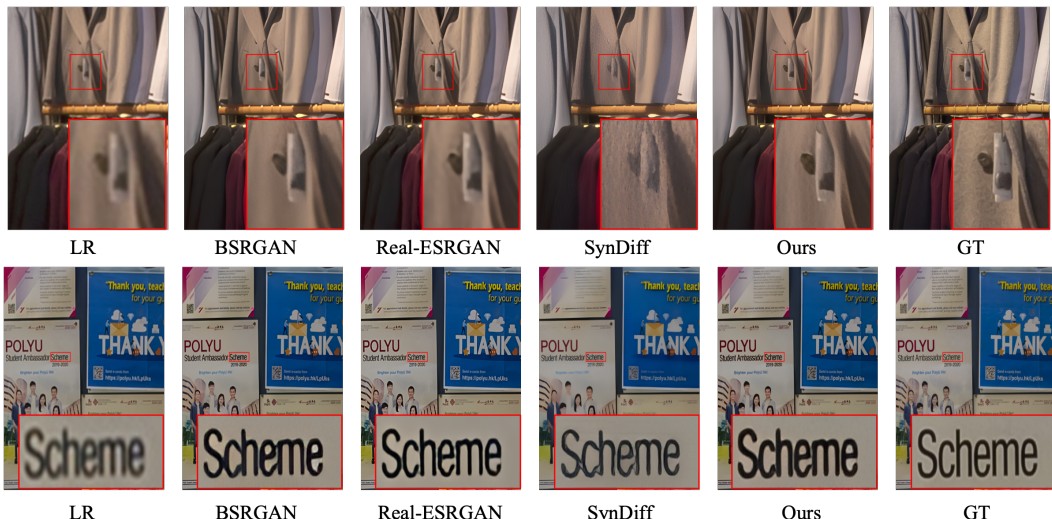

Figure 4: Visual comparison of Real SR based on different data generation methods. Real SR training using our data achieves the best visual results. Please zoom in for better visualization.

under six metrics, including distribution distance metric: FID, image structure and texture similarity metric: DISTS, and perceptual metric: LPIPS and CLIP-score, *etc*. It demonstrates the superiority of our Decoupled DDPM and verifies that our generation method is closer to target real LR images.

**Generalization Ability on Out-of-distribution Data.** Although we have collected about 152,000 real LR images for training, unseen scenarios often arise in real-world applications. To further explore our generalization capabilities on out-of-distribution data, we randomly extract 200 images from the unseen real-world video SR data (Yang et al., 2021) for evaluation, denoted as RealVSR. Then, we capture the degradation representing those images to generate the training data and compare it with existing methods on the SwinIR-L model, and the results are shown in Table 6. We can observe that our method still achieves the best performance on the out-of-distribution RealVSR benchmark, surpassing the existing state-of-the-art methods by 0.2214 and 0.0055 in PSNR and SSIM, respectively. This demonstrates the ability of our method to generalize for unseen scenarios in the real world.

| $T$ | 0-200 | 0-300 | 0-400 | 0-500 |
|---|---|---|---|---|
| PSNR↑ | 28.762 | **28.868** | 28.542 | 26.713 |
| SSIM↑ | 0.8501 | **0.8522** | 0.8435 | 0.7931 |

Table 4: Ablation study on $T$ in our proposed Decoupled DDPM during inferencing.

| Method | w/o $E_{deg}$ | w/o $E_{cont}$ | Ours |
|---|---|---|---|
| PSNR↑ | 21.642 | 25.745 | **28.868** |
| SSIM↑ | 0.7432 | 0.8095 | **0.8522** |

Table 5: Ablation study on degradation extractor $E_{deg}$ and content extractor $E_{cont}$.

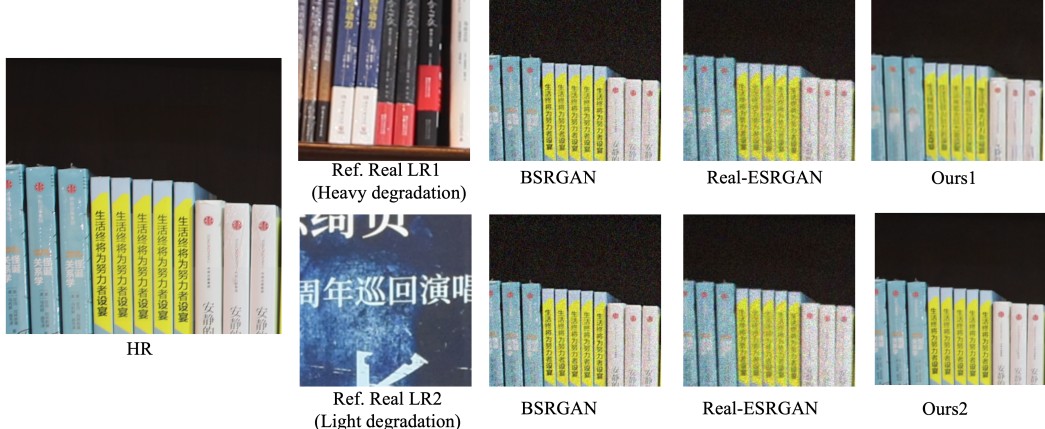

Figure 5: Visual comparison of generated low-resolution images with different reference real LR images. Please zoom in for better visualization.

## 3.3 QUALITATIVE RESULTS

We visualize the results of the generated LR images, as shown in Figure 3. It is clear that the Real-ESRGAN and BSRGAN, which are unable to perceive real degradation, exhibit a significant gap compared to real LR images. The SynDiff also struggles to adaptively capture the real degradation and content representation, resulting in unreal degradation distribution and low fidelity, as shown in the third row in Figure 3. However, benefiting from our adaptive perception ability of the degradation of real LR image and content of HR image, our RealDGen achieves the best visual effect in terms of both the realism of degradation and the fidelity of content. To demonstrate our method's ability to capture the degradation distribution of arbitrary LR images, we simulate data using degradation representations from various real-world LR images and content representation from an HR image (Figure 5). In contrast to existing methods like BSRGAN and Real-ESRGAN, which rely on handcrafted degradation models and struggle to accurately reflect degradation in specific LR images, our approach separates content representation from the HR image and extracts degradation representation from the reference LR image, enabling more accurate and realistic data generation.

To further present the effectiveness of our generated data for Real SR models, we visualize the super-resolved results of the SwinIR model on the RealSR and DRealSR benchmarks, as shown in Figure 4. Models trained on SynDiff, BSRGAN, and Real-ESRGAN data exhibit noticeable artifacts and blurring, shile our method consistently delivers superior visual quality without such defects. The textual scene in the second row of Figure 4 highlights the clear advantage of our approach. Additional visual comparisons are provided in Appendix A.16.

## 3.4 ABLATION STUDY

**Degradation and content extractor.** Our core idea is utilizing the degradation extractor $E_{deg}$ and content extractor $E_{cont}$ to extract degradation and content representation to control Decoupled DDPM in generating realistic LR. To verify this, we eliminate components $E_{deg}$ and $E_{cont}$, subsequently training the decoupled DDPM and fine-tuning the remaining extractor. For quick evaluation, we employ SwinIR-L as the baseline and evaluate on the Smartphone benchmark. The results in Table 5 demonstrate that the absence of $E_{deg}$ and $E_{cont}$ will cause mismatch and unreal degradation and content distortion problems and result in performance degradation, thereby affirming the indispensability of $E_{deg}$ and $E_{cont}$ in our method.

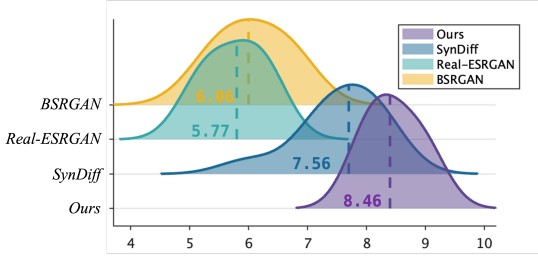

| RealVSR | PSNR↑ | SSIM↑ |
| --- | --- | --- |
| Real-ESRGAN | 22.7402 | 0.6939 |
| BSRGAN | 22.6564 | 0.6779 |
| SynDiff | 22.4607 | 0.6704 |
| Ours | **22.9616** | **0.6994** |

Figure 6: User study of generated real LR.

Table 6: The performance comparison with existing data generation method on out-of-distribution RealVSR benchmark.

$T$ **in data generation.** As illustrated in Section 2.3, in inferencing, we denoise from an initial LR image $\mathbf{x}_t$ with $t$-step noise added and perform t steps of denoising to improve the fidelity of Decoupled DDPM, where $t$ is selected from the range of 0 to $T$. To explore it, we set $T$ to 200, 300, 400, and 500 and conduct ablation experiments, as presented in Table 4. It is observed that the optimal $T$ is 300. A lower $T$ will result in insufficient generation of degradation, while a high $T$ will lead to a decrease in image fidelity. We propose that the value of $T$ should be dynamically adjusted to align with the degradation level of the given LR image, rather than being manually set. In future work, we plan to develop an automatic selection mechanism for $T$ to enhance the model's control and improve the fidelity of the generated data. More ablation studies of loss function's hyperparameters, $n$, and $margin$, are presented in Appendix A.7.

### 3.5 USER STUDY

To demonstrate the superiority of RealDGen in generating accurate and realistic low-resolution images, we conduct a user study involving 10 real-world LR images randomly chosen from existing benchmarks. 10 volunteers are asked to rate each scene individually (0: not similar at all; 2: not very similar; 4: slightly similar; 6: moderately similar; 8: similar; 10: extremely similar). Then, we aggregate the scores from all volunteers, and the results are shown in Figure 6. We can observe that the previous methods are unable to adaptively and accurately perceive real degradation, resulting in low fidelity in the generated real LR data, which leads to a general perception among users that there is a significant gap compared to real LR images. However, our RealDGen can adaptively capture the real degradation to accurately generate realistic LR images, resulting in an average score of 8.46 from human evaluators, surpassing previous approaches and thereby highlighting the improved visual quality that our method offers.

## 4 CONCLUSION

In this paper, we introduce a novel RealDGen to adaptively generate large-scale, high-quality paired data with arbitrary real LR as degradation reference and unpaired HR as content reference. Well-designed extractors and strategies are proposed to facilitate the extraction of robust content and degradation representations. A content-degradation decoupled diffusion model is proposed to adaptively generate realistic LR with given unpaired LR and HR conditions. Extensive experiments demonstrate that RealDGen not only achieves the best performance in generating realistic and accurate real LR images but also comprehensively improves the generalization ability of various popular SR models on real-world benchmarks. In addition, benefiting from the unsupervised learning of our method and the convenience of real LR image collection, it is easy to collect more real LR images with various real degradations to enhance generalization capability further.

**Limitation analysis.** Due to the high stochasticity inherent in diffusion models, RealDGen sometimes struggles to preserve fine textures. In future work, we plan to incorporate perceptual loss during training and develop a robust mechanism for automatically selecting the optimal denoising step based on the degradation level, thereby further enhancing the fidelity of the generated data.

**Acknowledgments.** This work was supported by the Natural Science Foundation of China under Grants 62225207,62436008 and 62206262.

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

# A APPENDIX

## A.1 TRAINING DETAIL AND ANALYSIS OF CONTENT AND DEGRADATION EXTRACTOR

### A.1.1 CONTENT EXTRACTOR

We initially contemplate leveraging the auto-regressive architecture of VQGAN Esser et al. (2021) as our Content Extractor $E_{cont}$ to capture content representations. However, our experimental endeavors reveal that the Generative Adversarial Network (GAN) in VQGAN impedes fine-tuning and the extraction of realistic content representations. Consequently, we elect to employ the encoder component of VQVAE Razavi et al. (2019) as $E_{cont}$. Given that these extractors have not been trained on degradation scenarios, they are not immediately suitable for our method. To surmount this challenge, we have meticulously crafted a fine-tuning strategy predicated on reconstruction learning. Specifically, we adopt Real-ESRGAN to degrade high-resolution (HR) images to generate paired datasets for training. During training, we maintain the decoder of VQVAE in a frozen state while fine-tuning the encoder, using HR images for supervised learning, with the objective loss function being the L2 loss. Our fine-tuning regimen is conducted with a learning rate of $1 \times 10^{-5}$, utilizing 8 NVIDIA V100 GPUs, which spent approximately one week for training. The reconstructed samples rendered by our fine-tuned Content Extractor are delineated in Figure 7. It is evident that after fine-tuning, VQVAE adeptly captures content representations in degraded scenarios and adeptly reconstructs high-resolution images.

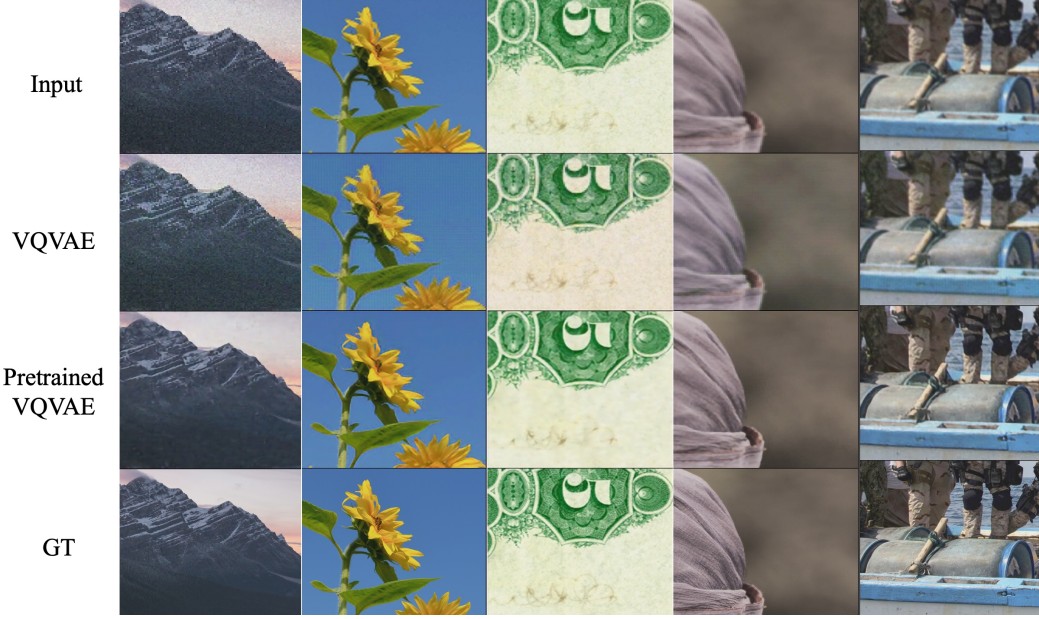

Figure 7: Visutal comparison with input LR image, VQVAE, our pretrained VQVAE, and GT. VQVAE, after training on our designed learning strategy, can better capture the robust content representations to reconstruct high-quilty HR.

**Why can the content feature extractor pre-trained on LR data robustly extract content features from HR data during data generation?** The proposed content extractor is designed to robustly extract content information from an arbitrary image. Given the inevitable presence of degradation in real-world HR images, we collect 152,000 real images with various types and degrees of degradation for unsupervised fine-tuning. Within this dataset, images with relatively low degradation levels exhibit a distribution closer to that of HR images. Fine-tuning the content extractor on these real images improves its generalization capability in real-world scenarios. Quantitative results (Tables 1, 2, and 5) and qualitative results (Figures 3, 4, and 5) validate that our content extractor can effectively extract robust content representations from any HR image to generate realistic and high-fidelity LR images.

**Why use LR-HR pairs to pre-train the content extractor?** The content extractor is designed to extract robust 'pure' content information from an image while ignoring potential degradation. Using LR-LR or HR-HR pairs causes the content extractor to learn identity mapping, mixing content, and degradation representations. During inference, we aim to extract content from a relatively clear image ('HR') without involving its degradation and combine it with the degradation of another image ('LR') to generate a new LR image. Therefore, we should use LR-HR pairs to train the content extractor. Table 2 and Figure 3 validate that our content extractor can effectively extract robust content representations to generate realistic and high-fidelity LR images.

### A.1.2  DEGRADATION EXTRACTOR

We introduce a novel Degradation Extractor denoted as $E_{deg}$, which is comprised of a feature extraction layer integrated with 16 residual blocks and a mapping function facilitated by adaptive polling and a 4-layer convolutional structure, as shown in Figure 8. Specifically, given the input of a low-resolution image into the Degradation Extractor, it engenders the output degradation representation $F_{deg} \in \mathbb{R}^{1 \times 1 \times 2048}$. To ensure the extraction of comprehensive and unique degradation representations in the LR image, we employ both reconstruction learning and contrastive learning methodologies in training our network. The training is executed on 8 NVIDIA V100 GPUs over the course of approximately seven days, with the learning rate configured at $1e - 4$.

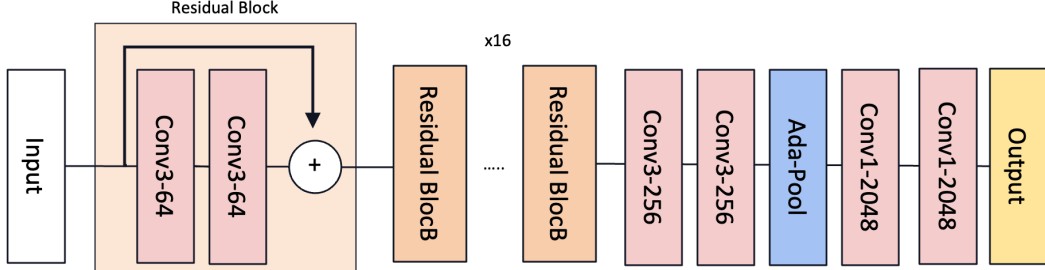

Figure 8: Illustration of our proposed degradation extractor.

### A.2  DETAIL OF HR AND LR RECONSTRUCTION NETWORK

To better adapt to the content extractor, that is, the encoder of the VQVAE, the decoder of VQVAE is adopted as HR reconstruction network $Rec_{hr}$ tasked with the reconstruction of high-resolution images. Given that the LR reconstruction network is tasked with reconstructing the LR image from content and degradation representations, we have adopted the aforementioned modulation network as our LR reconstruction network $Rec_{lr}$.

### A.3  TRAINING DETAILS OF DECOUPLE DDPM

During the training of the Decoupled DDPM, we configure the maximum diffusion step to be 500, span the training over 100 epochs, and utilize a learning rate of $1e - 4$ for the decoupled diffusion model. For the fine-tuning of the extractors, we apply a more refined learning rate of $1e - 6$. The batch size is defined as 8, and the entire training regimen is on 8 NVIDIA V100 GPUs, which typically consume around 14 days to complete.

We have amassed a collection of approximately 152,000 real low-resolution images sourced from publicly available datasets such as Wei et al. (2020); Cai et al. (2019); Ignatov et al. (2017) and those captured using smartphones. To elaborate, we have extracted all low-resolution images from these datasets and have cropped each to a uniform size of $256 \times 256$, resulting in a total of 110,000 images. Subsequently, to enhance the diversity of real-world degradation distribution, we have additionally procured 42,000 low-resolution images, each of the same $256 \times 256$ size. Collectively, this corpus of 152,000 low-resolution images serves as the training data for our Decoupled DDPM.

### A.4 DETAIL OF MODULATION BLOCK

As detailed in Section 2, we commence with real low-resolution images, denoted as $\mathcal{X}_{lr}$, which are derived from the real-world degradation distribution $q$. Utilizing our pre-trained extractors $E_{cont}$ and $E_{deg}$, we meticulously extract the respective content and degradation representations, $F_{cont}$ and $F_{deg}$. To authentically emulate the intricacies of the real imaging process, we introduce a modulation block, denoted as $\mathcal{M}$, which seamlessly integrates the degradation representation into the content representation. The integration is mathematically articulated as follows:

$$\mathbf{c} = \mathcal{M}\left(E_{deg}\left(\mathbf{x}_{lr}\right), E_{cont}\left(\mathbf{x}_{lr}\right)\right) \tag{8}$$

In this section, we proceed to elucidate the intricate mechanisms underpinning the modulation block $\mathcal{M}$. $\mathcal{M}$ is meticulously constructed from a series of four modulation layers, each comprising a convolutional layer, an activation function, and a modulation unit, as shown in Figure 9. The inputs to $\mathcal{M}$ encompass both the degradation and content representations. Within each modulation unit, the degradation representation is subjected to a sophisticated fusion process. Ultimately, the block culminates in its output through a final convolutional operation, synthesizing the enhanced representation.

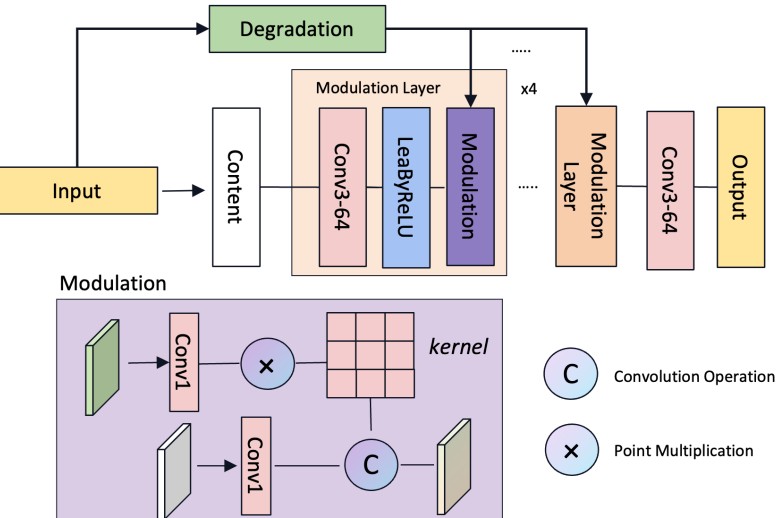

Figure 9: llustration of our modulation block.

### A.5 ANALYSIS AND DETAILS OF FINE-TUNING EXTRACTOR

During the second phase of training, we froze the majority of parameters in the content extractor, allowing only the first block within the VQVAE to undergo fine-tuning. Simultaneously, we applied a similar approach to the degradation extractor, restricting fine-tuning to the final convolutional layer. To validate the effectiveness of fine-tuning, we conduct an ablation experiment. In the second phase, we completely freeze the extractor and train only the Decoupled DDPM. We find that the realism of the generated degradation decreases because the extractor is trained exclusively on synthetic datasets, which results in its inability to extract real degradation representations and content. Consequently, the subsequent DDPM also has difficulty fitting the real LR accurately. Specifically, we test on the RealSR dataset using SwinIR-L, resulting in a 0.25dB performance degradation in PSNR with fine-tuning.

### A.6 STOCHASTICITY OF DIFFUSION

As discussed in Sec. 4, to mitigate the stochasticity of diffusion, we propose a filtering mechanism to eliminate outlier data. Specifically, we extract the degradation and content representations of the three simulated LR images, compare their similarities with the input LR's degradation and the HR's content representations, and select the best one to enhance the fidelity and quality of the generated LR images.

| RealSR | PSNR↑ | SSIM↑ | LPIPS↓ | FID↓ |
|---|---|---|---|---|
| w/o Filtering | 25.8965 | 0.7886 | 0.2316 | 116.8970 |
| Ours | **26.1615** | **0.7940** | **0.2226** | **107.2375** |

Table 7: The similarity results of ablation study on proposed filtering mechanic.

| SwinIR-L | PSNR↑ | SSIM↑ |
|---|---|---|
| w/o Filtering | 25.8747 | 0.7783 |
| Ours | **26.0250** | **0.7810** |

Table 8: The performance of SwinIR-L on RealSR benchmark.

To validate this, we conduct experiments on the RealSR dataset, calculating the similarity and data distribution compared to the real LR images in RealSR. Additionally, we verify the effectiveness using the downstream SR network SwinIR-L, with results presented in Tables 7 and 8. It can be observed that the data generated without the filtering mechanism exhibits lower similarity and further divergence from the real LR images, as well as reduced performance in the downstream SR network. This confirms the effectiveness and practicality of our proposed filtering mechanism in real-world scenarios.

## A.7 MORE ABLATION STUDIES

| $n$ | 1 | 0.1 | 0.01 | 0.001 | 0.0001 |
|---|---|---|---|---|---|
| PSNR↑ | 28.34 | 28.43 | 28.66 | 28.52 | 28.60 |
| FID↓ | 31.37 | 30.78 | 30.00 | 30.49 | 30.15 |

Table 9: Ablation studies on $n$.

| $margin$ | 1 | 2 | 3 | 4 | 5 |
|---|---|---|---|---|---|
| PSNR↑ | 28.35 | 28.50 | 28.66 | 28.61 | 28.60 |
| FID↓ | 30.97 | 30.61 | 30.00 | 30.16 | 30.15 |

Table 10: Ablation studies on $margin$.

In the loss function of the manuscript, we set the number of samples $n$ and $margin$ to 3 and 0.01, respectively, based on our experimental results. As shown in Tables 9 and 10, evaluating PSNR and FID between synthetic LR and real LR images on the DRealSR dataset, this configuration yields the best performance. Since it is impractical to exhaustively test all possible configurations, we empirically chose these values.

## A.8 MORE DOWNSAMPLE SCALE

Table 11: Comparison of data generation performance of different methods at different downsample scales on RealSR datasets.

| Downsample Scale | Metric | BSRGAN | Real-ESRGAN | SynDiff | Ours |
|---|---|---|---|---|---|
| ×4 | PSNR (dB)↑ | 23.26 | 24.39 | 25.24 | 26.16 |
| ×2 | PSNR (dB)↑ | 23.26 | 24.39 | 25.24 | 26.16 |

In our manuscript, we adopt the typical downsample scale ×4 scale setting to validate the effectiveness of the proposed data generation method. Furthermore, our method is easily adaptable to synthesizing images at different super-resolution scales, such as ×2, and ×4. To further demonstrate the flexibility and superiority of our approach, we present comparison results for ×4 and ×2. As shown in Table 11, our method achieves the best performance at both ×4 and ×2 scales.

## A.9 INFERENCE TIME AND PARAMETERS COMPARISON

**Inference time.** We test the average inference time (including IO, image processing, and generation) of producing 100 images and report the PSNR between synthetic LR and real LR on the RealSR dataset, as shown in Table 12. Using the official implementation, we measure inference time on 4 V100 GPUs. Our method adopts a sampling step of adding noise and denoising from 1 to 30, generating the highest-quality LR images and being faster than the learning-based SynDiff, although it is slower than RealESRGAN and BSRGAN. Our method and RealESRGAN can be used collaboratively and efficiently. We can use RealESRGAN for pre-training and then apply our method to create a small amount of high-quality data for fine-tuning the target scene, which does not require

Table 12: Comparison of inference time with different methods.

| Methods | Hand-Crafted Physical-Based | | Learning-Based | |
|---|---|---|---|---|
| | BSRGAN | Real-ESRGAN | SynDiff | Ours |
| Platform | 4 CPU | 4 V100 | 4 V100 | 4 V100 |
| Times (s) | 0.4267 | 0.1085 | 0.7801 | 0.6086 |
| PSNR (dB) | 23.26 | 24.39 | 25.24 | 26.16 |

Table 13: Parameters and performance comparison.

| RealSR | Param (M) | PSNR↑ | SSIM↑ | LPIPS↓ | FID↓ | DISTS↓ | CLIP-Score↑ |
|---|---|---|---|---|---|---|---|
| RealSR (Syndiff) | 101.121 | 25.24 | 0.7588 | 0.2371 | 119.950 | 0.1944 | 0.7984 |
| RealSR (Ours) | **73.676** | **26.16** | **0.7940** | **0.2226** | **107.2375** | **0.1865** | **0.8238** |
| DRealSR (Syndiff) | 101.121 | 28.0125 | 0.8136 | 0.1739 | 31.0855 | 0.1673 | 0.8462 |
| DRealSR (Ours) | **73.676** | **28.6629** | **0.8360** | **0.1497** | **30.0078** | **0.1567** | **0.8577** |
| SmartPhone (Syndiff) | 101.121 | 28.5020 | 0.8075 | 0.2423 | 70.6379 | 0.2044 | 0.8213 |
| SmartPhone (Ours) | **73.676** | **29.0054** | **0.8292** | **0.2121** | **69.1162** | **0.1964** | **0.8334** |

high speed from our method. For example, as shown in Table 2, fine-tuning ResShift on our generated realistic data significantly improves its generalization capability in real-world scenarios. Additionally, we will explore more efficient sampling strategies and distillation methods to speed up the inference of our method further.

**Model parameters.** Considering that BSRGAN and Real-ESRGAN are not deep-learning-based methods, following your suggestion, we compare the capacity and performance of deep learning-based Syndiff in Table 13. We see that our model not only has fewer parameters but also consistently achieves better performance on RealSR, DRealSR, and SmartPhone benchmarks.

## A.10 PERFORMANCE COMPARISON WITH REAL DATA COLLECTION

To further explore the performance comparison on real paired data, we further train the RRDB model using the real-collected training subset of RealSR and evaluate it on the test subset, as shown in Table 14. Since the training and test data in RealSR are captured using the same camera, they share a similar degradation distribution. Consequently, the model trained on real data achieves better performance compared to those trained on simulated data. However, collecting real-world data is often impractical, highlighting the importance of an accurate simulation method. Compared to other data synthesis approaches, our method is capable of generating realistic large-scale paired training data across diverse scenarios, enhancing the generalization capability of SR models. Furthermore, by incorporating our generated data into the real-collected training set, as shown in the last row of Table 14, the SR model's performance improves further, demonstrating the effectiveness of our approach.

## A.11 MORE ABLATION STUDIES ON DEGRADATION AND CONTENT EXTRACTORS

**Performance comparison before and after fine-tuning degradation and content extractor.** To enhance the generalization of $E_{deg}$ and $E_{cont}$ on unseen real distributions, as shown in the main text, only a small portion of the parameters in $E_{deg}$ and $E_{cont}$ are fine-tuned. We fine-tune the last layer, as shown in Figure 2. Therefore, it's nearly impossible for information to be directly passed through to the output. To further validate this, we conducted comparative experiments. As shown in Table 15, even without fine-tuning, our method can still generate realistic real LR, achieving the best performance compared to existing methods. After fine-tuning the extractors, the network's generalization ability in real scenarios is further enhanced, allowing for better generation of real LR in real-world settings.

**Validating the Effectiveness of Pretraining $E_{deg}$ and $E_{cont}$.** To Validate the effectiveness of the pretraining stage, we removed the pretrained weights of $E_{deg}$ and $E_{cont}$ and proceeded to train them alongside DDPM. However, without pretraining, $E_{deg}$ and $E_{cont}$ were unable to distinguish between

Table 14: Performance comparison with existing data synthesis methods and real data collection.

|  | PSNR↑ | SSIM↑ |
|---|---|---|
| RRDB (Real-ESRGAN) | 24.579 | 0.7614 |
| RRDB (BSRGAN) | 25.406 | 0.7685 |
| RRDB (SynDiff) | 25.488 | 0.7691 |
| RRDB (Ours) | 26.238 | 0.7747 |
| RRDB (Real Data) | 27.302 | 0.7934 |
| RRDB (Real Data+Ours) | 27.302 | 0.7934 |

Table 15: Performance comparison before and after fine-tuning degradation and content extractor.

| RealSR | PSNR↑ | SSIM↑ | LPIPS↓ |
|---|---|---|---|
| Real-ESRGAN | 24.39 | 0.679 | 0.388 |
| BSRGAN | 23.26 | 0.604 | 0.509 |
| SynDiff | 25.24 | 0.758 | 0.237 |
| Our methods w/o pretraining | 24.13 | 0.761 | 0.301 |
| Our methods w/o finetuning | 25.93 | 0.786 | 0.234 |
| **Our methods** | **26.16** | **0.794** | **0.222** |

content and degradation effectively. This incapacity led to a significant decline in performance when synthesizing data, as they failed to extract degradation accurately from Real LR. Simultaneously, the content extractor struggled to capture high-fidelity content representations, resulting in generated images with lower fidelity, as shown in Table 15. We observed that without pretraining, $E_{deg}$ and $E_{cont}$ could not properly extract the respective degradations, making it challenging to maintain content consistency and degradation authenticity during the generation process, leading to a marked drop in network performance.

## A.12 COMPARISON WITH BICUBIC

We incorporate bicubic degradation as a baseline and compare the SR performance. To ensure a fair comparison, we utilize the officially released pre-trained RRDB models trained on bicubic data and evaluate them on the RealSR and Smartphone datasets, as shown in Tables 16. The results clearly demonstrate that the RRDB model trained on our data achieves significantly better SR performance across various evaluation metrics, particularly in the perceptual-oriented LPIPS metric. This highlights the limitations of a single bicubic model, which struggles to handle the high complexity of real-world degradations (Wang et al., 2021b; Zhang et al., 2021).

## A.13 PERFORMANCE EVALUATION ON BLURRED LR SCENARIOS

To evaluate performance on blurred LR scenarios, we select blurred LR images from the popular REDS benchmark (Nah et al., 2019) to test and validate the improved generalization of our proposed method in more real-world scenarios. Specifically, we evaluate RRDB models pre-trained using synthesized data from different methods. The results, as shown in Table 17, demonstrate that our method consistently achieves the best performance on out-of-distribution data with blur degradation.

## A.14 COMPARISON WITH DIFFERENT CONTRASTIVE LEARNING METHODS

In (Wang et al., 2021a), the contrastive learning approach for training the degradation model constructs positive samples from different patches of the same image and negative samples from different images. However, the degradation between different patches of the same image is not always identical. For instance, in a defocused LR image, the defocused and non-defocused areas exhibit completely different degradations, and the degradation between different images may sometimes be similar. In contrast, our strategy constructs positive samples by keeping the degradation consistent while varying the content and generates negative samples by keeping the content consistent while varying the degradation, as described in Section 2.1 of the main text. This design ensures that the degradation

Table 16: Performance comparison with bicubic.

| Dataset | | PSNR↑ | SSIM↑ | LPIPS↓ |
|---|---|---|---|---|
| **Smartphone** | Bicubic | 28.45 | 0.845 | 0.395 |
| | Our method | **28.75** | **0.850** | **0.312** |
| **RealSR** | Bicubic | 25.98 | 0.755 | 0.442 |
| | Our method | **26.23** | **0.774** | **0.297** |

Table 17: Performance comparison on REDS.

| REDS | BSRGAN | Real-ESRGAN | Syndiff | Ours |
|---|---|---|---|---|
| PSNR↑ | 23.4629 | 23.6475 | 23.6472 | 23.9834 |
| SSIM↑ | 0.6615 | 0.6600 | 0.6443 | 0.6712 |

extractor captures the complete and unique degradation distribution of the current LR image. To validate this, we replace the degradation model in (Wang et al., 2021a) and train the diffusion model, then generate real LR images on the RealSR dataset. The results, shown in Table 18, demonstrate that our proposed degradation model produces more realistic LR images compared to (Wang et al., 2021a).

### A.15 MORE VISUAL COMPARISON ON INTERNET REAL LR IMAGES

To evaluate real LR images from the internet, we compared existing methods on the SwinIR model and visualized the results, as shown in Figure 10 in the Appendix. It can be seen that our method better generates textures, while other methods tend to be over-smooth, validating the effectiveness of our approach.

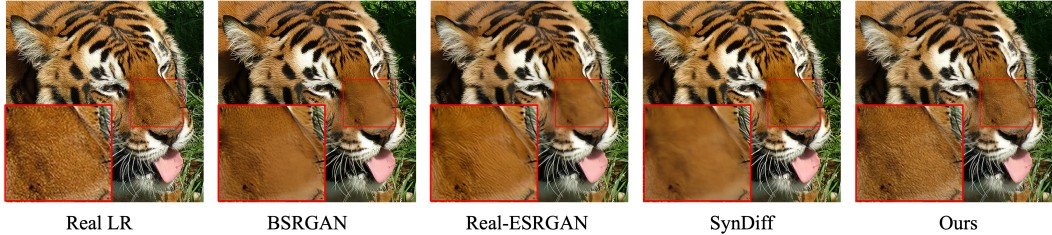

| Real LR | BSRGAN | Real-ESRGAN | SynDiff | Ours |

Figure 10: Visual comparison of generated HR image from real internet LR images.

### A.16 MORE VISUAL COMPARISON

Here, we present additional visual results on the DRealSR benchmark to demonstrate the superiority of our method in adaptively generating accurate and realistic LR images, as shown in Figure 11 and 12. Additionally, we also present some visual results on the SmartPhone benchmark, as shown in Figure 13.

### A.17 MORE RELATED WORK

Recent and notable advancements in generative models and real-world image super-resolution, low-level vision tasks can be explored in these papers (Lu et al., 2024a; 2023; 2024b; Peng et al., 2025; Li et al., 2023; Lugmayr et al., 2020b; Li et al., 2022a; Peng et al., 2024b; Sun & Chen, 2024; Peng et al., 2024a; Lugmayr et al., 2019a; Peng et al., 2024c; Di et al., 2024; Wang et al., 2023; Peng et al., 2020; Chen et al., 2023a; Huang et al., 2020; Lugmayr et al., 2020b; Li et al., 2022a; Lugmayr et al., 2019a; Sun & Chen, 2024; Miao et al., 2024; Li et al., 2024b; Wang et al., 2022; 2024a;b; Han et al., 2024a;b;a; Ye et al., 2025; 2023; Zhang et al., 2025b; Gu et al., 2023; 2024; He et al., 2025b;a; 2024; Zhang et al., 2025a; 2023a; Li et al., 2024a).

Table 18: Performance comparison with different contrastive learning.

| RealSR | PSNR↑ | SSIM↑ | LPIPS↓ |
|---|---|---|---|
| DASR | 25.73 | 0.778 | 0.265 |
| Our method | 26.16 | 0.794 | 0.222 |

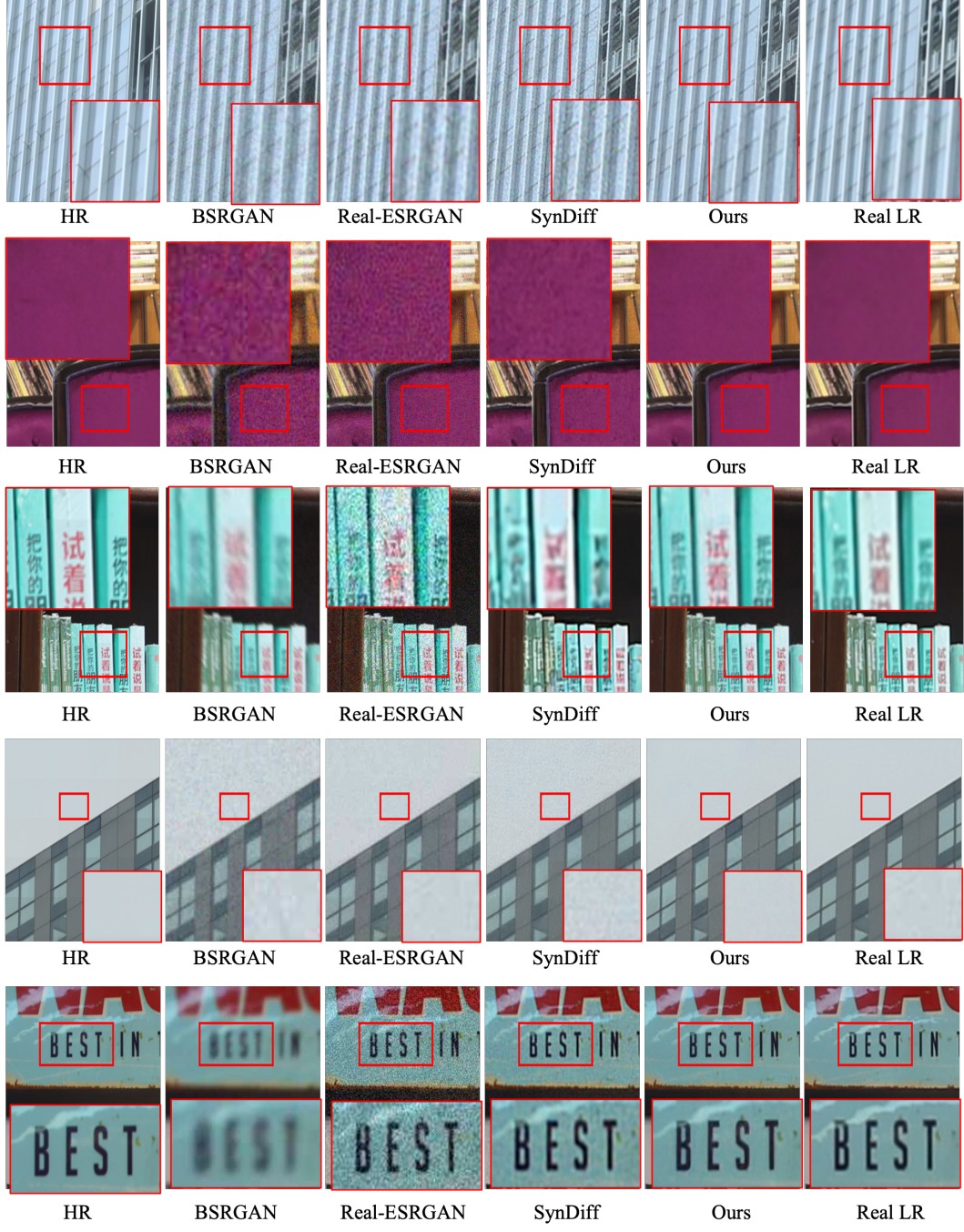

Figure 11: Visual comparison of generated LR on DRealSR.

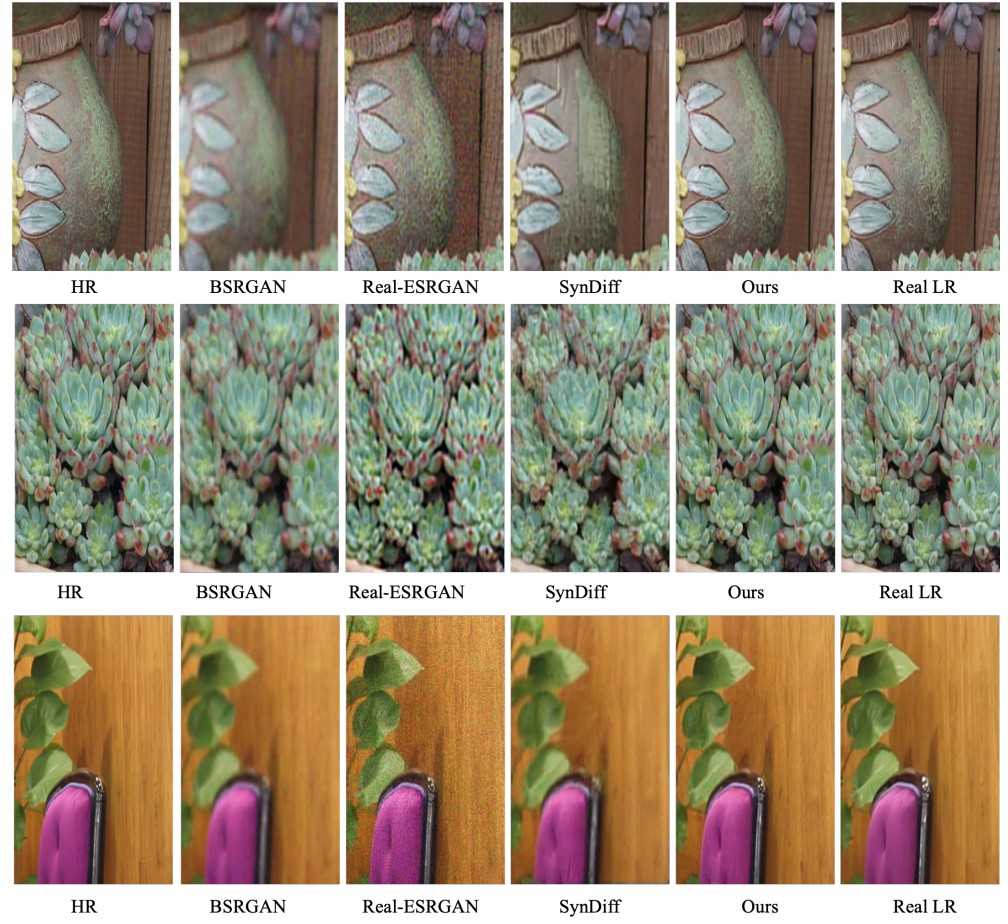

Figure 12: Visual comparison of generated LR on DRealSR.

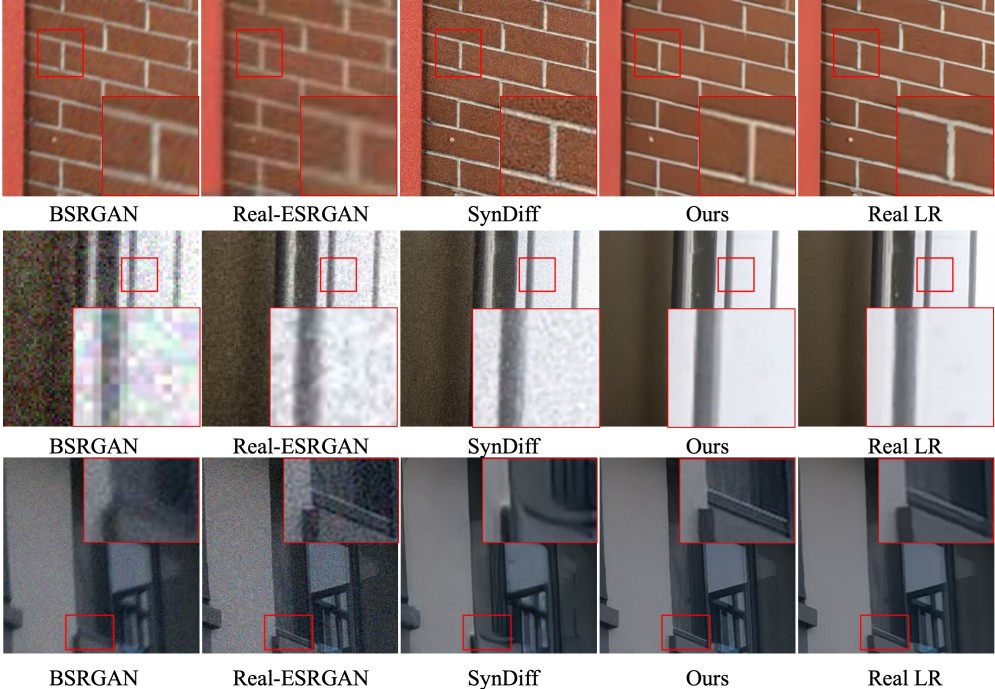

Figure 13: Visual comparison of generated LR images on the smartphone benchmark.

