# OpenReview forum: "Towards Realistic Data Generation for Real-World Super-Resolution"
_ICLR.cc/2025/Conference — ICLR 2025 Poster_

### Official Review · Reviewer_X8hR · 2024-10-27

**Soundness:** 3
**Presentation:** 3
**Contribution:** 3
**Rating:** 5
**Confidence:** 5

**Summary:**

In this approach, a generative model is introduced to synthesize realistic low-resolution images, which are subsequently used as a training dataset to improve the performance of conventional super-resolution networks. In particular, unlike traditional methods, the proposed generative model is trained in an unsupervised manner using an unpaired training dataset, thus eliminating the dependency on high-quality ground truth images during the inference phase-a novel aspect of this approach. Furthermore, experimental results show that the generated low-resolution images exhibit improved accuracy compared to those produced by conventional methods, leading to superior performance in downstream tasks such as super-resolution.

**Strengths:**

This work introduces a novel generative model that eliminates the need for paired high-resolution and low-resolution images captured in real-world scenarios. Instead, by utilizing a synthetically generated training dataset from RealESRGAN, the proposed model effectively trains two key modules, E_deg and E_cont, which are designed to separate degradation and content in the given low-resolution (LR) images. This allows the model to generate realistic LR images using only a single LR input image, without the need for a corresponding high-resolution ground truth, setting it apart from conventional approaches. Given that collecting a large volume of real-world paired datasets is both time-consuming and costly, the proposed method, which bypasses the use of such paired real-world datasets, offers a significant advantage over traditional methods. Moreover, experimental results quantitatively and qualitatively demonstrate the superior performance of the generative model, further validating its efficacy compared to traditional approaches.

**Weaknesses:**

Although the proposed method removes the reliance on paired real-world datasets for both the training and inference phases, it still has limitations in fully demonstrating its performance. Specifically, the work lacks comprehensive experiments and detailed analysis. For further clarification, please refer to the questions outlined below.

Minor point:
In line 87, the method by Park et al. is incorrectly described as a GAN-based approach. It actually utilizes normalizing flows for generation, and this should be revised accordingly.

**Questions:**

1.	The proposed method seems to share several ideas with Syndiff (Yang et al.). Could you please clarify the key contributions and differences between the two approaches?
2.	To train the E_deg and E_cont modules, the paper utilizes paired synthetic images generated by RealESRGAN. However, real-world low-resolution (LR) images can vary significantly from these synthetic images, which may hinder the accurate separation of degradation and content, potentially leading to suboptimal super-resolution (SR) performance on external datasets, as observed in Table 6. Could you clarify whether the proposed method is limited by the degradation distribution of RealESRGAN?
3.	Additionally, E_deg and E_cont are fine-tuned during the training of ReadDGen. As a result, real-world LR images presented at test time could bypass these modules and be directly passed through to the output. It would be useful for the authors to explain how these LR images differ from the ground truth (GT) LR images, and to demonstrate that the proposed generator can synthesize diverse LR samples. Moreover, more results before and after fine-tunign would be also beneficial.
4.	What would happen if a single, non-pretrained encoder were used instead of the E_deg and E_cont modules? Could you discuss the impact of this change to see whether separation of content and degradation is necessary?
5.	The SR results presented in the tables are somewhat lower than expected. Were these SR networks trained for 4x super-resolution (SR)? Please specify the scale factor. Additionally, it would be helpful to include more SR results using bicubic LR images for the baseline, as well as SR results trained on real-world SR datasets as oracles.
6.	While the adaptability of the proposed method is emphasized in the paper, the evidence provided to support this claim seems insufficient. It would be beneficial for the authors to include additional analysis and results to further substantiate this claim.

---

> ### Author Response · Authors · 2024-11-20
> **Response to Reviewer X8hR (Part1)**
>
> Thank you very much for recognizing our work: "**novel generative model**", "**offers a significant advantage over traditional methods**" and "**superior performance**". Next, we will address each of your concerns.
>
> **Minor Point:** Thank you for your correction. We have updated this in the revised version at lines 86-87.
>
> **1: Differences to Syndiff**
>
> On the one hand, Syndiff is limited to generating degradation distributions similar to the training data, which restricts its ability to generalize to novel and diverse LR domains. This limitation makes it less suitable for handling various real-world LR scenarios. In contrast, our method adaptively extracts degradation representations from real LR images and generates corresponding paired data. This adaptability enhances the SR model's generalization across diverse scenarios, as demonstrated in Tables 1 and 3 of the main text.
>
> On the other hand, Syndiff's generation process lacks explicit content and degradation conditions to control the output, often resulting in content loss and unrealistic degradation. Our method addresses these issues by extracting degradation representations from real LR images and content from HR images, using these as conditions to guide the diffusion process. This approach ensures better control over both content fidelity and degradation realism. As shown in the first row of Figure 4 in the main text, Syndiff-generated LR images suffer from color distortion, content loss, and unrealistic degradation. In contrast, our method produces images with significantly higher content fidelity and degradation characteristics that closely match real LR images, thereby demonstrating its superiority in both realism and accuracy.
>
> **2: Degradation modeling beyond RealESRGAN degradations**
>
> Our training process is conducted in two stages. In the initial pre-training stage, Real-ESRGAN is utilized to provide the degradation extractor with an initial capacity for degradation representation extraction. However, the degradation distribution learned at this stage lacks sufficient realism for diverse real-world scenarios. To address this, we fine-tune a subset of the degradation extractor's parameters using real LR images. This fine-tuning process adapts the learned degradation distribution to better align with real-world LR images, significantly enhancing the extractor's generalization ability. The effectiveness of this approach is supported by our experimental results. As shown in Table 3 of the main text, our method consistently generates more realistic LR images across various real-world settings, including RealSR, DRealSR, and Smartphone datasets. For instance, our generated data achieves 26.16 dB on RealSR, compared to 24.40 dB from Real-ESRGAN. This improved realism translates into superior SR model performance, as evidenced by the results in Tables 1 and 2. Additionally, we provide visualizations of synthesized LR images to further validate our method. As shown in Figure 3, LR images synthesized by Real-ESRGAN exhibit noticeable discrepancies compared to real LR images, whereas our method produces LR images with much higher similarity to real data. Figure 5 also validates that our method can extract different degradation representations from real-world LR images, enabling more accurate and realistic data generation. These findings demonstrate that our approach effectively models degradation distributions beyond the capabilities of Real-ESRGAN.

---

> ### Author Response · Authors · 2024-11-20
> **Response to Reviewer X8hR (Part2)**
>
> **3: Performance comparison before and after fine-tuning for degradation and content encoders.**
>
> To enhance the generalization of E_deg and E_cont on unseen real-world distributions, we fine-tune a small subset of their parameters. Specifically, we only fine-tune the last layers of E_deg and E_cont, as depicted in Figure 2 of the main text. This design ensures that information cannot bypass the former layers directly to the final output, preserving the integrity of the learned representations. To further validate this point, we conduct additional experiments. As shown in Table R1, even without fine-tuning, our method generates highly realistic LR images and achieves superior performance compared to existing methods. After fine-tuning, the network's generalization ability in real-world scenarios is further enhanced, enabling more effective synthesis of LR images under diverse conditions. Moreover, as illustrated in Figure 5 of the main text, our method can generate LR images with varying degradation levels by extracting degradation representations from different LR images while preserving the content representation from an HR image. We greatly appreciate your valuable feedback and have added a detailed illustration in Appendix A.11 of the revised version.
>
> **Table R1: Performance Comparison of LR Synthesis on RealSR before and after Fine-tuning**
>
> |               Methods               | PSNR$\uparrow$ | SSIM$\uparrow$ | LPIPS$\downarrow$ |
> | ---------------------------------- | -------------- | -------------- | ----------------- |
> |             Real-ESRGAN             |      24.39      |      0.679      |        0.388        |
> |                BSRGAN                |      23.26      |      0.604      |        0.509        |
> |               SynDiff               |      25.24      |      0.758      |        0.237        |
> |     Our methods w/o fine-tuning     |    25.93    |    0.786    |      0.234      |
> | **Our methods w fine-tuning** | **26.16** | **0.794** |   **0.222**   |
>
>
> **4: Effectiveness of Pre-training E_deg and E_cont**
>
> Following your suggestion, we remove the pre-trained weights of E_deg and E_cont and train them jointly with DDPM from scratch. However, without pre-training, E_deg and E_cont fail to effectively distinguish between content and degradation. This limitation prevents the degradation extractor from accurately extracting degradation information from real LR images, leading to a significant decline in the accuracy of synthesized LR images. Similarly, the content extractor struggles to capture high-fidelity content representations, resulting in generated images with reduced fidelity, as demonstrated in Table R2. These results highlight the critical role of pre-training in establishing the initial capacity of E_deg and E_cont, enabling them to effectively separate and extract content and degradation information. We have added this comparison in Appendix A.11 of the revised manuscript.
>
> **Table R2: Performance comparison of LR synthesis on RealSR between with and without pre-training of E_deg and E_cont**
>
> |           Methods           | PSNR$\uparrow$ | SSIM$\uparrow$ | LPIPS$\downarrow$ |
> | -------------------- | -------------- | -------------- | ----------------- |
> |      w/o pre-training      |      24.13      |      0.761      |        0.301        |
> | **with pre-training** | **26.16** | **0.794** |   **0.222**   |

---

> ### Author Response · Authors · 2024-11-20
> **Response to Reviewer X8hR (Part3)**
>
> **5: Comparison with bicubic LR images.**
>
> Following your suggestion, we incorporate bicubic degradation as a baseline and compare the SR performance. To ensure a fair comparison, we utilize the officially released pre-trained RRDB models trained on bicubic data and evaluate them on the RealSR and Smartphone datasets, as shown in Tables R3 and R4. The results clearly demonstrate that the RRDB model trained on our data achieves significantly better SR performance across various evaluation metrics, particularly in the perceptual-oriented LPIPS metric. This highlights the limitations of a single bicubic model, which struggles to handle the high complexity of real-world degradations [2,3]. Thank you for your valuable feedback. We have added this comparison to Appendix A.12 of the revised manuscript.
>
> **Table R3: Comparison with RRDB model trained on bicubic data on RealSR benchmark**
>
> |     RealSR     | PSNR$\uparrow$ | SSIM$\uparrow$ | LPIPS$\downarrow$ |
> | ------------ | -------------- | -------------- | ----------------- |
> |    Bicubic    |      25.98      |      0.755      |        0.442        |
> | **Ours** |      26.23      |      0.774      |        0.297        |
>
> **Table R4: Comparison with RRDB model trained on bicubic data on Smartphone benchmark**
>
> |   Smartphone   | PSNR$\uparrow$ | SSIM$\uparrow$ | LPIPS$\downarrow$ |
> | ------------ | -------------- | -------------- | ----------------- |
> |    Bicubic    |      28.45      |      0.845      |        0.395        |
> | **Ours** |      28.75      |      0.850      |        0.312        |
>
> [2] Real-ESRGAN: Training Real-World Blind Super-Resolution with Pure Synthetic Data
>
> [3] Designing a Practical Degradation Model for Deep Blind Image Super-Resolution
>
> **6: The Adaptability of the Proposed Method**
>
> Our method adaptively generates accurate and realistic paired data tailored to the target LR scenario. To validate its effectiveness, we conduct experiments on three real-world benchmarks: RealSR, DRealSR, and Smartphone. For each benchmark, we extract degradation representations specific to the LR scenario and apply them to the data generation process. Compared to existing methods, our approach consistently achieves the highest-quality data synthesis, enabling SR models to deliver superior performance across all benchmarks, as shown in Tables 1, 2, and 3 in the main text. Additionally, to evaluate the generalization capability of our method on out-of-distribution (OOD) data, we test it on the Real Video SR dataset, as detailed in lines 423–431 and Table 5 of the main text. Our method also demonstrates superior performance on this dataset, further validating its adaptability and robustness across diverse real-world scenarios.
>
> Thank you very much for your valuable suggestions. If you have any further suggestions, please do not hesitate to share them with us.

---

> ### Author Response · Authors · 2024-11-25
> **Looking Forward to Your Response, Dear Reviewer X8hR**
>
> Dear Reviewer X8hR,
>
> Thank you very much for recognizing our work: "**novel generative model**", "**offers a significant advantage over traditional methods**" and "**superior performance**".
>
> We would like to thank you again for the valuable time you devoted to reviewing our paper. Since the end of the discussion period is getting close and we have not heard back from you yet, we would appreciate it if you kindly let us know of any other concerns you may have and if we can be of any further assistance in clarifying them.
>
> Thank you once again for your contribution to the development of our paper.
>
> Authors

---

> > ### Comment · Reviewer_X8hR · 2024-11-27
> >
> > I appreciate the authors' time and effort in preparing the rebuttal. Many of my concerns have been addressed; however, a few questions remain. While the authors discuss the differences between the proposed method and SynDiff, these differences remain unclear and somewhat ambiguous. Please provide a more detailed explanation and clarify where the proposed method achieves the most significant improvements over SynDiff. Additionally, please specify how the results presented in Table R1 and Table R2 were obtained, including details about the datasets, baseline SR networks, and other relevant settings. Furthermore, I could not find the SR result of the proposed method (26.16 dB) in the main manuscript. Thank you.

---

> > > ### Author Response · Authors · 2024-11-27
> > > **Response to Reviewer X8hR**
> > >
> > > Thank you for your response. We are pleased to hear that most of your concerns have been addressed. We would like to further clarify the remaining questions:
> > >
> > > 1. **Differences to SynDiff**
> > >
> > > - **(1) Adaptability to New Real LR Domains**: Our method is highly flexible and can adapt to new real LR domains without re-training, unlike SynDiff. Our model features a robust degradation encoder that extracts degradation representations from arbitrary real LR images and applies them to HR images to synthesize new LR images. This allows our method to quickly adapt to new degradation domains. In contrast, SynDiff's sole input is the HR image, with degradation patterns inherent in the diffusion model determined by the training data, making it unsuitable for new LR domains. For example, as shown in Table 6, our method generates more realistic new-domain LR images on the OOD RealVSR benchmark, achieving a 0.5dB improvement over SynDiff.
> > > - **(2) High Fidelity Maintenance**: Our method maintains high fidelity between the original HR and generated LR images, a feat SynDiff cannot achieve. SynDiff initially uses Real-ESRGAN degradation to generate an LR image, then adds noise and performs denoising. The Real-ESRGAN degradation process leads to information loss, resulting in lower content fidelity of the generated LR images compared to the HR image, as shown in Figure 3 of the main text. In contrast, we propose a robust content encoder that extracts content representations from the clear HR image. This content representation is used as a condition during the denoising process of the diffusion model, thereby improving content fidelity, as illustrated in Figure 3.
> > >
> > > 2. **Detailed Explanation of Table R1 and Table R2**
> > >
> > > - **(1) Table R1**: This table demonstrates the effectiveness of fine-tuning the degradation and content encoders (after pre-training). We generate LR images using encoders with and without fine-tuning and compare their accuracy with real LR images on the RealSR dataset, as detailed in lines 377–422 of the main text. Table R1 shows that fine-tuning the encoders significantly improves generation accuracy, producing LR images that more closely align with real LR images across PSNR, SSIM, and LPIPS metrics compared to other methods.
> > >
> > > - **(2) Table R2**: To equip our encoders with degradation and content extraction capabilities, we pre-train them on synthesized paired data before using them to train the diffusion model on the real unpaired data. Without pre-training, meaning training the encoders from scratch with the diffusion model, the encoders cannot effectively separate degradation and content, resulting in the loss of adaptability and high fidelity. Table R2 compares the accuracy of generated LR images with real LR images on the RealSR dataset, using models with and without encoder pre-training. The results indicate that pre-training enables the encoders to effectively disentangle and extract content and degradation information, thereby enhancing the realism of the generated LR images.
> > >
> > > 3. **Location of 26.16dB in the Main Text**
> > >
> > > The value 26.16dB represents the PSNR result of our method when evaluating the LR generation accuracy on the RealSR dataset, as shown in **Table 3 of the main text (26.1615dB)**. For brevity, this value is approximated to 26.16dB. To enhance clarity, we will update this value in the revised version.
> > >
> > > Thank you again for your valuable feedback.

---

> ### Author Response · Authors · 2024-12-01
> **Looking Forward to Your Response, Dear Reviewer X8hR**
>
> Dear Reviewer X8hR,
>
> Thank you for your valuable contribution to the development of our paper. **We are pleased to hear that most of your concerns have been addressed**. As the end of the discussion period is approaching, we look forward to your response. We would appreciate it if you could let us know of any remaining concerns or if there is any further assistance we can provide to clarify them.
>
> Best regards,
>
> Authors

---

### Official Review · Reviewer_vCs5 · 2024-10-31

**Soundness:** 2
**Presentation:** 3
**Contribution:** 2
**Rating:** 6
**Confidence:** 4

**Summary:**

The authors present a new framework, Realistic Decoupled Data Generator (RealDGen), to generate realistic, large-scale data for real-world image super-resolution. The framework includes a contrastive learning-based degradation extractor and a reconstruction-based content extractor, enabling the effective capture of authentic degradation patterns and content features from real-world data. Utilizing the pre-trained extractors, the authors decouple the degradation and content features of given real low-resolution (LR) images, which are subsequently used as conditioning inputs for a Decoupled Diffusion Probabilistic Model (DDPM) to reconstruct the given images. Finally, by leveraging unpaired LR and HR datasets, the trained DDPM is able to generate realistic LR images, which exhibit degradation patterns closely resembling those found in real-world data. Extensive experiments across various SR models demonstrate that RealDGen consistently enhances SR performance on multiple real-world benchmarks. Detailed comments are listed below.

**Strengths:**

1. The proposed RealDGen framework effectively separates content and degradation features using a diffusion model, facilitating the generation of realistic low-resolution images that more accurately replicate real-world degradation patterns.
2. By utilizing a well-designed contrastive learning approach for degradation extraction and a reconstruction-based method for content extraction, the proposed RealDGen framework effectively decouples degradation and content features, thereby enhancing data realism and adaptability across different models.
3. Extensive experiments on multiple real-world SR benchmarks show that RealDGen consistently enhances the performance of various SR models, highlighting its practical effectiveness for real-world applications.

**Weaknesses:**

1. What is the technical contribution of the proposed method RealDGen? It seems that the effectiveness of the proposed methods mainly comes from the powerful diffusion model, and a similar idea of separating the degradation and content features has been investigated by previous methods [A].
2. During the training phase of DDPM, the authors finetune partial parameters of the extractor. However, it is unclear the motivation and the effect of finetuning partial parameters. Please provide more discussions.
3. The proposed method is tested on several SR benchmarks, but more experiments on diverse data with various types of degradation (e.g., motion or defocus blur [B, C]) would better showcase the framework’s generalization ability and robustness in a wider range of real-world scenarios.
4. The proposed RealDGen relies on a large real LR dataset, which may be a potential limitation for scenarios where collecting such data is difficult or infeasible. It would be useful to investigate the effect of different amounts of real LR data on data generation.
5. The authors use a contrastive learning approach for the degradation extractor but do not provide enough detail on how different contrastive learning configurations (e.g., different negative sampling strategies) could affect the performance of data generation. It would be better to conduct more additional ablations to investigate the effect of different contrastive learning methods, such as [D].
6. Since the proposed method requires training a DDPM from scratch, the data generation cost is substantially high. Please provide a cost comparison with other existing methods to facilitate a clearer understanding of the computational demands and efficiency of the proposed approach.

[A] Learning Many-to-Many Mapping for Unpaired Real-World Image Super-resolution and Downscaling, TPAMI 2024.
[B] Benchmarking Neural Network Robustness to Common Corruptions and Perturbations, ICLR 2019.
[C] Efficient Test-Time Adaptation for Super-Resolution with Second-Order Degradation and Reconstruction, NeurIPS 2023.
[D] Unsupervised Degradation Representation Learning for Blind Super-Resolution, CVPR 2021.

**Questions:**

What is the technical contribution of the proposed method RealDGen?

---

> ### Author Response · Authors · 2024-11-20
> **Response to Reviewer vCs5 (Part1)**
>
> Thank you very much for recognizing our work: "**facilitating the generation of realistic low-resolution images that more accurately replicate real-world degradation patterns**", "**enhancing data realism and adaptability across different models**" and "**highlighting its practical effectiveness for real-world applications**".
>
> Next, we will address each of your concerns.
>
> **1. Differences to method [A]**
>
> First, the data synthesis approach in [A] generates degradation distributions limited to the training data, often failing to generalize effectively to new and diverse LR domains. In contrast, our method extracts degradation representations directly from real LR images of new domains, enabling the production of accurate LR images and demonstrating significantly stronger generalization capabilities.
>
> Second, the decoupling process in [A] is based on two key assumptions:
> a. The assumption that LR and HR images share the same content representation. However, in scenarios with high downsampling rates and severe degradations, the content of LR images can undergo substantial changes, violating this assumption.
> b. The assumption of a linear hypothesis in the representation space, expressed as $Z_d = Z_{lr} - Z_c$, where degradation representation $Z_d$ is obtained by subtracting the content representation $Z_c$ from the LR feature $Z_{lr}$. This hypothesis assumes a linear relationship in the latent space, while deep representations do not inherently satisfy such linearity.
>
> In contrast, our method avoids these strict assumptions by employing carefully designed contrastive and reconstruction strategies to learn unique and complete degradation representations as well as pure content representations. This approach ensures greater practicality and robustness in real-world scenarios. As the code for [A] is unavailable and the rebuttal time is limited, we plan to reproduce this method and include comparative experiments in the final version of the paper.
>
> **2. Why is it necessary to fine-tune the partial parameters of the extractors?**
>
> After pre-training with well-designed strategies using synthetic LR images, our extractors effectively capture robust content and degradation representations. To further improve the generalization of the extractors on unseen real distributions, we fine-tune a subset of their parameters, as illustrated in Figure 2 and discussed in lines 213–215 of the main text. To validate the effectiveness of this fine-tuning, we conduct comparative experiments on the RealSR dataset, with and without fine-tuning, as presented in Table R1. The results clearly show that fine-tuning enhances the model's ability to capture real degradation and content features, resulting in more realistic synthesized LR images. We appreciate your valuable suggestion and have included this analysis in Appendix A.11 of the revised manuscript.
>
> **Table R1: Performance comparison of LR synthesis on RealSR between with and without fine-tuning**
>
> |          Method          | PSNR$\uparrow$ | SSIM$\uparrow$ | LPIPS$\downarrow$ |
> | :----------------------: | :--------------: | :--------------: | :-----------------: |
> |     w/o fine-tuning     |      25.93      |      0.786      |        0.234        |
> | **w fine-tuning** | **26.16** | **0.794** |   **0.222**   |
>
> **3. Performance evaluation on blurred LR scenarios.**
>
> Following your suggestion, we select blurred LR images from the popular REDS benchmark [1] to test and validate the improved generalization of our proposed method in more real-world scenarios. Specifically, we evaluate RRDB models pre-trained using synthesized data from different methods. The results, as shown in Table R2, demonstrate that our method consistently achieves the best performance on out-of-distribution data with blur degradation. Thank you for your valuable suggestion, and we have included this analysis in Appendix A.13 of the revised manuscript.
>
> **Table R2: Performance comparison on REDS.**
>
> | Method      | PSNR$\uparrow$ | SSIM$\uparrow$ |
> | ----------- | ---------------- | ---------------- |
> | BSRGAN      | 23.4629          | 0.6615           |
> | Real-ESRGAN | 23.6475          | 0.6600           |
> | Syndiff     | 23.6472          | 0.6443           |
> | **Ours**       | **23.9834**         | **0.6712**           |
>
> [1] NTIRE 2019 Challenge on Video Deblurring and Super-Resolution: Dataset and Study

---

> ### Author Response · Authors · 2024-11-20
> **Response to Reviewer vCs5 (Part2)**
>
> **4. Performance comparison with different quantities of Real LR images**
>
> In the training process, considering that real LR images are relatively easy to collect, we utilize real LR images from public datasets and self-collected data to ensure that the generation model fully learns the degradation distribution of real-world scenarios. However, during testing, only a small number of real LR images captured in the test scenarios are required to generate realistic paired data. For instance, in RealSR, we use only 100 real LR images for data synthesis, which is relatively easy to obtain, and achieves high-quality results, as shown in Table 3 of the main text. To further evaluate the performance of the proposed method with fewer real LR images, we conduct comparative experiments by reducing the number of real LR images to 50 and testing the performance of the RRDB model, as presented in Table R3. The results show that, compared to existing methods, our approach significantly enhances the generalization ability of the RRDB model in real-world scenarios, achieving the best performance.
>
> **Table R3: Performance comparison with existing data synthesis methods**
>
> | Method             | PSNR$\uparrow$ | SSIM$\uparrow$ |
> | ------------------ | ---------------- | ---------------- |
> | RRDB (Real-ESRGAN) | 24.740           | 0.762            |
> | RRDB (BSRGAN)      | 25.245           | 0.766            |
> | RRDB (SynDiff)     | 25.233           | 0.767            |
> | **RRDB (Ours)**        | **26.203**           | **0.771**            |
>
> **5. Comparison with contrastive learning method [D]**
>
> In DASR [D], the contrastive learning approach for training the degradation model constructs positive samples from different patches of the same image and negative samples from different images. However, the degradation between different patches of the same image is not always identical. For instance, in a defocused LR image, the defocused and non-defocused areas exhibit completely different degradations, and the degradation between different images may sometimes be similar. In contrast, our strategy constructs positive samples by keeping the degradation consistent while varying the content and generates negative samples by keeping the content consistent while varying the degradation, as described in Section 2.1 of the main text. This design ensures that the degradation extractor captures the complete and unique degradation distribution of the current LR image. To validate this, we replace the degradation model in [D] and train the diffusion model, then generate real LR images on the RealSR dataset. The results, shown in Table R4, demonstrate that our proposed degradation model produces more realistic LR images compared to [D]. Additional ablation studies on our contrastive learning strategy are presented in Appendix A.7. Thank you for your suggestion. We have included this comparison in Appendix A.14 of the revised version.
>
> **Table R4: Performance comparison with difference contrastive learning**
>
> |     Method     | PSNR$\uparrow$ | SSIM$\uparrow$ | LPIPS$\downarrow$ |
> | ------------ | -------------- | -------------- | ----------------- |
> |    DASR [D]    |      25.73      |      0.778      |        0.265        |
> | **Ours** | **26.16** | **0.794** |   **0.222**   |
>
> **6. Training Time**
>
> Our model contains only 73M parameters and requires approximately 768 V100 GPU hours (4 days on 8 V100 GPUs) for training. In comparison, Syndiff has 101M parameters and requires around 960 GPU V100 hours (5 days on 8 V100 GPUs). Additionally, unlike Syndiff, our method needs to be trained only once and can be adaptively applied to different real LR scenarios without requiring retraining, offering both efficiency and flexibility.
>
> **7. Related Papers**
>
> Thank you for your advice. We have included these related papers [A-D] in the revised version.
>
> [A] Learning Many-to-Many Mapping for Unpaired Real-World Image Super-resolution and Downscaling, TPAMI 2024.
>
> [B] Benchmarking Neural Network Robustness to Common Corruptions and Perturbations, ICLR 2019.
>
> [C] Efficient Test-Time Adaptation for Super-Resolution with Second-Order Degradation and Reconstruction, NeurIPS 2023.
>
> [D] Unsupervised Degradation Representation Learning for Blind Super-Resolution, CVPR 2021.
>
> Thank you very much for your valuable suggestions. If you have any further suggestions, please do not hesitate to share them with us.

---

> > ### Comment · Reviewer_vCs5 · 2024-11-22
> >
> > Thanks for the detailed response from the authors. The responses have addressed most of my concerns, so I have decided to raise the score to recommend accepting the paper.

---

> > > ### Author Response · Authors · 2024-11-22
> > > **Thanks**
> > >
> > > We appreciate your recognition of our work and are grateful for the time you took to provide feedback.

---

### Official Review · Reviewer_3zDX · 2024-11-01

**Soundness:** 3
**Presentation:** 2
**Contribution:** 4
**Rating:** 8
**Confidence:** 5

**Summary:**

The paper presents a novel approach for generating realistic data for real-world image super-resolution (SR) tasks, addressing the limitations in the generalization of SR models due to the divergence between training data and real-world degradations. The authors propose a new framework called Realistic Decoupled Data Generator (RealDGen), which leverages unsupervised learning to decouple content and degradation from real low-resolution (LR) and high-resolution (HR) images. This decoupling is integrated into a diffusion model, allowing the generation of paired training data that more accurately reflects real-world conditions. The paper claims that RealDGen significantly improves the generalization ability of SR models across various backbone architectures.

**Strengths:**

1. The paper identifies the challenges in SR, particularly the gap between the synthetic degradations used in training data and the degradations in real-world images. By positioning the problem within the existing literature, the authors provide a strong motivation for their approach.
2. The core idea of the paper lies in decoupling content and degradation in real-world images to generate realistic data for SR, which is a novel approach for this task. The proposed approach, RealDGen, provides a scalable and adaptive solution for generating large-scale, realistic datasets.
3. The authors conduct comprehensive evaluation of RealDGen across multiple SR models on different datasets. The results show consistent improvements in both PSNR-oriented and perceptual-oriented SR models, demonstrates the robustness of the proposed approach across a variety of settings.

**Weaknesses:**

1. The paper does not clearly define Real-world LR. Through experiments, we can see that the work is more focused on the SR problem in real-world photography, but the authors do not clearly define or explain it in the paper. This makes Figure 2 (b) somewhat difficult to understand.
2. The overall presentation of this paper shoude be improved. The symbols in the formula do not correspond well to the figures. Eq.5 is not reflected in Figure 2. The colors used in Figure 1 (b) is somewhat confusion. The representation of 'G', 'S' in Figure 1 (a) should be 'N', 'K'. X_hr in Eq.5 is not explained. Line 179, 'for an HR image'.

**Questions:**

How far is the performance of the model trained using the simulated data from that of the model trained using real collected data?

---

> ### Author Response · Authors · 2024-11-20
> **Response to Reviewer 3zDX**
>
> Thank you very much for recognizing our work: "**provide a strong motivation**", "**a novel approach**" and "**the robustness of the proposed approach across a variety of settings**".
>
> Next, we will address each of your concerns.
>
> **1. Clarifying Real-World LR in Figure 2 (b) and Providing More Visual Comparisons on Real LR Images from the Internet**
>
> Real-world LR images originate from various sources, including photography (e.g., images captured by smartphones and DSLRs), compressed low-quality images from the Internet, and other scenarios. To evaluate the effectiveness of our method comprehensively, we test on widely used benchmarks such as RealSR and DRealSR, as well as our collected images captured by smartphones. As demonstrated in Tables 1, 2, and 3 of the main paper, our method consistently achieves superior results. Additionally, to assess the generalization capability of our method on Internet-derived LR images, we provide visualizations in Figure 10 (Appendix A.15). These examples show that our method reconstructs much clearer textures compared to other approaches. We greatly appreciate this valuable feedback and have incorporated a detailed description in Appendix A.15 of the revised manuscript.
>
> **2. Minor Presentation Details**
>
> a. Mismatch between Figure 2 and Eq. 5
>
> Thank you for pointing this out. We have updated Figure 2 accordingly.
>
> b. Inconsistent Colors in Figure 1
>
> Thank you for your suggestion. We have aligned the colors of different existing methods and their corresponding cycles in Figure 1 to enhance the readability of the paper.
>
> c. Minor Issue in Figure 1 (a)
>
> We have aligned the representations of "G," "S," "N," and "K" in Figure 1. The updated Figure 1 clearly indicates these elements.
>
> d. Explanation of X_{hr}
>
> X_{hr} in Eq.5 refers to HR images. We have added the relevant description in the revised version at 194-196 lines.
>
> **3. Performance Comparison with Real Collection Data**
>
> **Table R1: Performance comparison with existing data synthesis methods and real data collection**
>
> | Methods                 | PSNR$\uparrow$ | SSIM$\uparrow$ |
> | ----------------------- | ---------------- | ---------------- |
> | RRDB (Real-ESRGAN)      | 24.579           | 0.7614           |
> | RRDB (BSRGAN)           | 25.406           | 0.7685           |
> | RRDB (SynDiff)          | 25.488           | 0.7691           |
> | RRDB (Ours)             | 26.238           | 0.7747           |
> | RRDB (Real Data)        | 27.302           | 0.7934           |
> | **RRDB (Real Data + Ours)** | 27.413           | 0.7956           |
>
> Following your suggestion, we further train the RRDB model using the real-collected training subset of RealSR and evaluate it on the test subset, as shown in Table R1. Since the training and test data in RealSR are captured using the same camera, they share a similar degradation distribution. Consequently, the model trained on real data achieves better performance compared to those trained on simulated data. However, collecting real-world data is often impractical, highlighting the importance of an accurate simulation method. Compared to other data synthesis approaches, our method is capable of generating realistic large-scale paired training data across diverse scenarios, enhancing the generalization capability of SR models. Furthermore, by incorporating our generated data into the real-collected training set, as shown in the last row of Table R1, the SR model's performance improves further, demonstrating the effectiveness of our approach. We appreciate your valuable suggestion and have added this description to Appendix A.10 of the revised manuscript.
>
> If you have any suggestions, please do not hesitate to share them with us.

---

> > ### Comment · Reviewer_3zDX · 2024-11-22
> >
> > I appreciate the idea of this paper. After carefully reading the author's rebuttal and other reviews, I decide to maintain my Accept rating.

---

> > > ### Author Response · Authors · 2024-11-22
> > > **We sincerely appreciate your recognition and support.**
> > >
> > > We sincerely appreciate your recognition and support.

---

### Official Review · Reviewer_qCkN · 2024-11-04

**Soundness:** 2
**Presentation:** 3
**Contribution:** 3
**Rating:** 5
**Confidence:** 4

**Summary:**

This paper introduces an unsupervised learning framework designed to generate realistic LR images for real-world SR. This paper utilizes a decoupled approach for content and degradation extraction, incorporating them into a diffusion model to produce paired data that closely represents real-world degradations. Experimental results indicate that the proposed method improves the generalization and performance of SR models across various real-world benchmarks.

**Strengths:**

- The paper is well-structured, making the methodology and findings easy to understand.
- The paper achieves competitive SR results under an unpaired setting, which is practical and advantageous in real-world applications.

**Weaknesses:**

- The paper argues that content and degradation can be decoupled by content and degradation extractor.
However, it is unclear how the content extractor’s encoder is guaranteed to capture only pure content information.

- Equation (5) utilizes X_{hr}, but no explanation or justification is provided for its use in the main manuscript.

- The paper would benefit from a comparison of the total parameters of RealDGen with those of other methods, as this would provide additional context on scalability.

**Questions:**

Please refer to the Weaknesses section.

---

> ### Author Response · Authors · 2024-11-20
> **Response to Reviewer qCkN**
>
> Thank you very much for recognizing our work: "**improves the generalization and performance of SR models across various real-world benchmarks**", "**achieves competitive SR results**" and "**practical and advantageous in real-world applications**".  It is widely recognized that the quality of training data is crucial to model performance. To address the limitations of existing methods, we develop an innovative approach to produce large-scale, high-quality data that accurately simulate real-world degradations. This significantly improves the generalization capability of SR methods in real-world applications.
>
> Next, we will address each of your concerns.
>
> **1. Why does the content extractor extract "pure" content information?**
>
> The ability of the content extractor to capture 'pure' content information is primarily driven by the network's supervision objectives. The input to the content extractor is an LR image with various degradations, while the target is an HR image free of these degradations. From the perspective of model optimization, degradation information does not contribute to minimizing the objective function, which focuses on reconstructing high-resolution content. For example, the MSE loss emphasizes pixel-level accuracy, steering the model away from learning degradation-related features. Consequently, the network learns to disregard degradation characteristics and instead prioritize extracting 'pure' content information. As illustrated in Figure 7 of the appendix, our content extractor effectively separates degradation from content and extracts 'pure' content information, recovering high-quality results.
>
> **2. Explanation of X_{hr}**
>
> In Equation (5), X_{hr} refers to high-resolution images. We have clarified this in the revised manuscript on lines 194–196, highlighted in blue. Thank you for this valuable suggestion, which has helped improve the clarity of our work.
>
> **3. Parameters comparison**
>
> Considering that BSRGAN and Real-ESRGAN are not deep-learning-based methods, following your suggestion, we compare the capacity and performance of deep learning-based Syndiff in Table R1. We see that our model not only has fewer parameters but also consistently achieves better performance on RealSR, DRealSR, and SmartPhone benchmarks. Thank you very much for your valuable suggestions. We have included this comparison in section A.9 of the appendix in the revised version.
>
> **Table R1: Parameters and Performance Comparison**
>
> |           RealSR           | Parameters (M)$\downarrow$ | PSNR$\uparrow$ | SSIM$\uparrow$ | LPIPS$\downarrow$ | FID$\downarrow$  | DISTS$\downarrow$ | CLIP-Score$\uparrow$ |
> | ------------------------- | -------------------------- | --------------- | -------------------------  | -------------------------  | -------------------------  | -------------------------  | -------------------------  |
> |      RealSR (Syndiff)      |           101.121           |       25.2461       | 0.7588           | 0.2371              | 119.9950           | 0.1944              | 0.7984                 |
> |   RealSR (**Ours**)   |       **73.676** |  **26.1615** | **0.7940** | **0.2226** | **107.2375** | **0.1865** | **0.8238** |
> |      DRealSR (Syndiff)      |           101.121           |      28.0125      | 0.8136           | 0.1739              | 31.0855            | 0.1673              | 0.8462                 |
> |  DRealSR (**Ours**)  |       **73.676** | **28.6629** | **0.8360** | **0.1497** | **30.0078** | **0.1567** | **0.8577** |
> |    SmartPhone (Syndiff)    |           101.121           |      28.5020      | 0.8075           | 0.2423              | 70.6379            | 0.2044              | 0.8213                 |
> | SmartPhone (**Ours**) |       **73.676** | **29.0054** | **0.8292** | **0.2121** | **69.1162** | **0.1964** | **0.8334** |
>
> If you have any further suggestions, please do not hesitate to share them with us.

---

> > ### Author Response · Authors · 2024-11-25
> > **Looking Forward to Your Response, Dear Reviewer qCkN**
> >
> > Dear Reviewer qCkN,
> >
> > Thank you very much for recognizing our work: "**improves the generalization and performance of SR models across various real-world benchmarks**", "**achieves competitive SR results**" and "**practical and advantageous in real-world applications**".
> >
> > We would like to thank you again for the valuable time you devoted to reviewing our paper. Since the end of the discussion period is getting close and we have not heard back from you yet, we would appreciate it if you kindly let us know of any other concerns you may have and if we can be of any further assistance in clarifying them.
> >
> > Thank you once again for your contribution to the development of our paper.
> >
> > Authors

---

> > > ### Comment · Reviewer_qCkN · 2024-11-28
> > >
> > > Thank you for your detailed response.
> > >
> > > Most of my concerns have been addressed, but I still have further questions regarding the first concern.
> > >
> > > I think the content extractor is guaranteed to extract 'pure' content only under in-distribution degradation domains.
> > > For instance, in the denoising task, which is part of the real-world SR task, a denoising network trained on synthetic noise struggles to handle real-world noise,
> > > where out-of-distribution noise remains in the input (as shown in Figures 8, 9, and 10 of [1]).
> > >
> > > Even if the content extractor is fine-tuned in a second stage with unseen degradation domains, the target is reconstructed LR images.
> > > At this point, the content extractor may no longer extract 'pure' content within its embedding space,
> > > as the focus shifts to the reconstruction of the LR image rather than clean content.
> > >
> > > To summarize the questions:
> > >
> > > 1. How can the content extractor be guaranteed to extract 'pure' content under out-of-distribution degradation domains?
> > > 2. If the degradation and content extractors are fine-tuned on the same target (reconstructed LR images),
> > > can this process still be considered 'decoupled,' in line with one of the main contributions of the paper?
> > >
> > > [1] Masked Image Training for Generalizable Deep Image Denoising, CVPR 2023

---

> ### Author Response · Authors · 2024-11-28
> **Response to Reviewer qCkN**
>
> Thank you for your response. We are pleased to hear that most of your concerns have been addressed. We would like to further clarify the remaining questions:
>
> **1. Function of the content extractor**
>
> We need to clarify the function of the content extractor within our proposed framework. As illustrated in Figure 1(c), during the data generation process (inference period), we utilize the content extractor exclusively to process **high-resolution (HR) images** (sourced from DIV2K) to enhance the fidelity of the generated LR images. The input images are guaranteed to be high-quality and free from out-of-domain (OOD) degradations. On the other hand, in the second training stage depicted in Figure 2, if we can collect diverse real-world noisy images (as you suggested in [1]) for training, our content extractor can be fine-tuned to become more robust in extracting relatively pure content information.
>
> We acknowledge the challenge of extracting completely pure content information from various degraded images without training on such data. This is an area we plan to explore further, as it holds significant potential for improving our system's ability to process a more diverse range of images. We will cite [1] and add these discussions in the revised manuscript.
>
> [1] Masked Image Training for Generalizable Deep Image Denoising, CVPR 2023
>
> **2. The decoupling of content and degradation extractors**
>
> During the pre-training stage, the content and degradation extractors are trained on synthesized paired data using carefully designed strategies to acquire the ability to extract content and degradation representations, respectively, as described in Section 2.1 of the main text. In the second fine-tuning phase, where only LR images are available, we freeze most layers of the content and degradation extractors, leaving only the last layer of each extractor trainable, as shown in Figure 2 and lines 212-214 of the main text. This ensures that the main functionalities of the extractors remain largely unchanged, preserving the decoupling property.
>
> Table R1: Performance comparison of swapping degradation and content extractors.
>
> |          Methods          | PSNR$\uparrow$ | SSIM$\uparrow$ | LPIPS$\downarrow$ |
> | ----------------------- | --------------- | -------------- | ----------------- |
> |          w swap          |      7.8231      |      0.1533      |       0.9187       |
> | **w/o swap (Ours)** | **26.1615** | **0.7940** |  **0.2229**  |
>
> To verify this, we swap the two extractors during data generation to test if they become homogenized. We compare the generated LR images with real LR images from the RealSR dataset, with the accuracy reported in Table R1 below. The results show that swapping the extractors leads to a complete collapse in model performance, indicating their distinct roles in data generation. In contrast, our proposed method achieves excellent data generation accuracy, validating the effectiveness of the decoupling approach. We will include these discussions in the revised manuscript to enhance the readability of the paper.
>
>
> Thank you again for your valuable feedback.

---

> ### Author Response · Authors · 2024-12-01
> **Looking Forward to Your Response, Dear Reviewer qCkN**
>
> Dear Reviewer qCkN,
>
> Thank you for your valuable contribution to the development of our paper. **We are pleased to hear that most of your concerns have been addressed**. As the end of the discussion period is approaching, we look forward to your response. We would appreciate it if you could let us know of any remaining concerns or if there is any further assistance we can provide to clarify them.
>
> Best regards,
>
> Authors

---

### Author Response · Authors · 2024-11-20
**Response to AC and reviewers**

Dear AC and all reviewers,

We sincerely appreciate your time and efforts in reviewing our paper. We are glad to find that reviewers recognized the following merits of our work:

* **Novel approach [X8hR,3zDX]**:  As widely recognized, the quality of training data is crucial to network performance. To address the limitations of existing methods in adaptively generating realistic paired data, we introduce a novel unsupervised content-degradation decoupled diffusion model to generate large-scale, realistic, and diverse paired training data for real-world super-resolution. The idea is novel, and the paper is solid.
* **Impressive performance [X8hR,3zDX,qCkN,vCs5]**: With our proposed data generation method, the SR performance of various super-resolution networks, including CNN-based, Transformer-based, GAN-based, and diffusion-based models, is significantly enhanced across diverse real-world benchmarks.
* **Good robustness and **generalization** [3zDX,qCkN]** : Existing super-resolution methods face challenges in generalizing to real out-of-domain data. Our approach addresses this by extracting degradation representations from the target real LR domain, enabling rapid adaptation of SR methods to new domains and enhancing the performance of SR models across diverse real benchmarks and scenarios.

We also thank all reviewers for their insightful and constructive suggestions, which help further improve our paper. In addition to the pointwise responses below, we summarize the major revision in the rebuttal according to the reviewers’ suggestions:

* **Detailed illustration of the training process in Sec.2 [X8hR,3zDX,vCs5]**: We have provided a more detailed analysis and description, and added comparative experiments with other contrastive learning methods.
* **Additional compared methods [X8hR,3zDX]**: We have involved more models trained with bicubic training data and real-world training data for comparisons. The results show that our method achieves excellent performance.
* **Manuscript update [3zDX,qCkN,X8hR]**: We have included more in-depth parameter analysis, experimental results, and discussions in the main paper and Appendix.

We hope our pointwise responses below can clarify all reviewers' confusion and address the raised concerns. We thank all reviewers' efforts and time again.

Best,

Authors

---

### Meta-Review · Area_Chair_674q · 2024-12-25

**Metareview:**

This paper proposes a contrastive learning method to generate data for realistic image super-resolution in an unpaired setting. The image content and the degradation are decoupled to generate realistic and diverse LR images at scale.  The decoupling is guided by the learning objective, and demonstrated by ablation study.

The reviewers find the idea to be novel and the experimental results to be impressive.

The authors are advised to add the analysis provided in the rebuttal in the manuscript to reflect the feedback from the reviewers for better exposition.

**Additional Comments On Reviewer Discussion:**

Most of the concerns were addressed in the discussion from the author responses.

There was a discussion regarding how the content & degradation decoupling happens and how it could be verified. The authors provided experimental justification that decoupling actually happens.

Clarification on the difference from Syndiff was requested in the discussion with **vCs5** and **X8hR**. The authors explained the technical differences in the algorithm, the parameter requirements, and the domain adaptability.

---

### Decision · Program_Chairs · 2025-01-22

Accept (Poster)